# Fat body-specific reduction of CTPS alleviates HFD-induced obesity

Jingnan Liu[1,2†], Yuanbing Zhang[1,3,4†], Qiao-Qi Wang[1,3,4†], Youfang Zhou[1,3,4], Ji-Long Liu[1,5*]

[1]School of Life Science and Technology, ShanghaiTech University, Shanghai, China; [2]College of Life Sciences, Shanghai Normal University, Shanghai, China; [3]Institute of Biochemistry and Cell Biology, Shanghai Institutes for Biological Sciences, Chinese Academy of Sciences, Shanghai, China; [4]University of Chinese Academy of Sciences, Beijing, China; [5]Department of Physiology, Anatomy and Genetics, University of Oxford, Oxford, United Kingdom

**Abstract** Obesity induced by high-fat diet (HFD) is a multi-factorial disease including genetic, physiological, behavioral, and environmental components. *Drosophila* has emerged as an effective metabolic disease model. Cytidine 5'-triphosphate synthase (CTPS) is an important enzyme for the de novo synthesis of CTP, governing the cellular level of CTP and the rate of phospholipid synthesis. CTPS is known to form filamentous structures called cytoophidia, which are found in bacteria, archaea, and eukaryotes. Our study demonstrates that CTPS is crucial in regulating body weight and starvation resistance in *Drosophila* by functioning in the fat body. HFD-induced obesity leads to increased transcription of CTPS and elongates cytoophidia in larval adipocytes. Depleting CTPS in the fat body prevented HFD-induced obesity, including body weight gain, adipocyte expansion, and lipid accumulation, by inhibiting the PI3K-Akt-SREBP axis. Furthermore, a dominant-negative form of CTPS also prevented adipocyte expansion and downregulated lipogenic genes. These findings not only establish a functional link between CTPS and lipid homeostasis but also highlight the potential role of CTPS manipulation in the treatment of HFD-induced obesity.

## Editor's evaluation

This study describes a role for CTPS (Cytidine 5'-triphosphate synthase) and CTPS filamentous structures (cytoophidia) in regulating fat storage in the fly fat body in normal and high-fat diets. The data were collected and analyzed using validated, solid methodologies. These results are useful for biologists interested in general cellular mechanisms of metabolism.

**\*For correspondence:**
liujl3@shanghaitech.edu.cn

†These authors contributed equally to this work

**Competing interest:** The authors declare that no competing interests exist.

## Introduction

Obesity has been a worldwide epidemic disease for decades, characterized by the accumulation of excessive or abnormal amounts of fat, which poses a significant threat to health. The consumption of high-fat diets (HFDs) is a leading contributor to obesity, a major risk factor for chronic disorders such as diabetes, cardiovascular diseases, and cancer, responsible for approximately 2.5 million deaths yearly (*Ogden et al., 2007*). Understanding the mechanism of obesity and its related secondary diseases, such as nonalcoholic fatty liver, requires careful consideration of the harmful effects of genetic factors and excessive dietary fat consumption (*Pelusi and Valenti, 2019*). However, the precise interactions between genetic predisposition, environmental factors, and lifestyle factors in the etiology of obesity remain to be elucidated fully.

**eLife digest** The high rate of obesity has created a global health burden by leading to increased rates of chronic diseases like diabetes and cardiovascular disease. Tackling this issue is complicated as it is influenced by many factors, including genetics, behaviour and environment. To better understand the biochemical changes that underlie metabolic issues in a simpler setting, scientists can study fruit flies in the laboratory. These insects share many genes with humans and have similar responses to a high-fat diet.

Previous research identified an enzyme, called CTP synthase (CTPS), which is produced in large amounts by the liver and fat tissue in mammals, and the equivalent in fruit flies, known as the fat body. Multiple CTPS molecules can combine to form long strands of protein called cytoophidia, which have been seen in organisms ranging from humans to bacteria. Recent results showed that the fruit fly equivalent of CTPS drives fat cells to stick together, which is necessary to maintain and form fat tissue. However, it is not clear if altering the levels of CTPS can affect the response to a high-fat diet.

To address this, Liu, Zhang, Wang et al. studied fruit flies on a high-fat diet, showing that this increased the production of CTPS. When the flies were treated to deplete levels of CTPS in the fat body, they had less body weight gain, smaller fat cells and lower amounts of fats in the body. Genetically modified flies with a version of CTPS that was unable to form cytoophidia also showed fewer signs of obesity, indicating how the enzyme might influence the response to dietary fats.

These findings further implicate CTPS in the cause of obesity and help to understand its role. However, it remains to be seen if this also applies to humans. If this is the case, drugs that block the activity of CTPS could help to reduce the impact of a high-fat diet on public health.

CTPS is a rate-limiting enzyme in the de novo synthesis of CTP, catalyzing the transfer of amide nitrogen from glutamine to the C-4 position of UTP, a process that requires ATP (*Lieberman, 1955*; *Levitzki and Koshland, 1969*). In 2010, our research group and others reported that CTPS could polymerize into filamentous structures, termed cytoophidia (*Liu, 2010*) or CTPS filaments (*Ingerson-Mahar et al., 2010*; *Noree et al., 2010*). These structures were observed in fruit flies (*Liu, 2010*), bacteria (*Ingerson-Mahar et al., 2010*), and yeast cells (*Noree et al., 2010*), and subsequent research demonstrated their presence in human cells (*Chen et al., 2011*; *Carcamo et al., 2011*), plants, and archaea (*Daumann et al., 2018*; *Liu, 2011*; *Liu, 2016*; *Zhou et al., 2020*), indicating that CTPS filamentation is a highly conserved process across prokaryotic and eukaryotic organisms. Various studies have revealed that CTPS cytoophidia have several functions, including modulation of enzymatic activity (*Aughey et al., 2014*; *Aughey and Liu, 2015*), maintenance of cell morphology (*Ingerson-Mahar et al., 2010*), and stabilization of CTPS protein (*Liu, 2016*; *Sun and Liu, 2019*). Notably, CTPS cytoophidia have been found in several human cancers, including hepatocellular carcinoma (*Chang et al., 2017*). The precise role of CTPS cytoophidia in the development of these diseases is yet to be established; it is believed, however, that they play a role in maintaining tissue homeostasis by regulating cell growth, proliferation, and nutrient availability. It is worth noting that mammalian adipose or hepatic tissues produce a substantial amount of CTPS. Despite this, the specific physiological function of CTPS in lipid homeostasis remains an area of ongoing investigation.

*Drosophila* has emerged as a powerful and simplified model of metabolic diseases such as HFD-induced obesity, diabetes, and cardiovascular disease (*Birse et al., 2010*; *Oldham, 2011*; *Liu et al., 2012*; *Smith et al., 2014*) because it offers the opportunity to investigate the links between genetics, diet, and metabolism. Our previous investigation revealed that CTPS cytoophidia are abundantly distributed in various tissues, including the central nervous system, fat body and intestine of *Drosophila* (*Zhang et al., 2020*). The *Drosophila* fat body, an organ with high metabolic activity and conserved signaling pathways, plays a crucial role in sensing nutritional conditions and responding through the integration of lipid metabolism, acting as an equivalent to the mammalian liver or adipose tissue (*Li et al., 2019*; *Arrese and Soulages, 2010*). Our recent research has revealed that the single *Drosophila* ortholog of CTPS, which forms cytoophidia in larval adipocytes, promotes adipocyte adhesion mediated by integrin-Collagen IV (*Zhang et al., 2020*; *Liu et al., 2022*).

Using the *Drosophila* model, we discovered a significant physiological function for CTPS in lipid metabolism and metabolic adaptation following HFD exposure. Our study revealed that HFD feeding

results in an upregulation of CTPS transcription and an elongation of CTPS cytoophidia in larval adipo-cytes. Our findings provide in vivo evidence that CTPS depletion prevents body weight gain and restricts adiposity. These results suggest that adipocytes utilize CTPS to regulate lipid metabolism and adapt to metabolic changes, which may have implications for developing metabolic disease.

## Results

### Fat body-specific knockdown of CTPS leads to body weight loss

We employed *Drosophila* as a model organism to explore the potential involvement of CTPS in fat deposition and obesity. To achieve this, we used different drivers to knock down the expression of CTPS specifically in various tissues. First, we globally knocked down the expression of *CTPS* using a ubiquitous temperature-sensitive driver, *TubG4ts* (*tubulin-GAL4, tubulin-GAL80ts*), and cultured flies at 25 °C. Female flies in the *TubG4ts*>*CTPS*-Ri group (1.003 mg; S.E.M.: ±0.056 mg) weighed 16.4%, 14.4%, and 11.0% less than those in the *TubG4ts*> + (1.2 mg; S.E.M.: ±0.066 mg), *CTPS*-Ri/+ (1.172 mg; S.E.M.: ±0.079 mg), and *TubG4ts*>Con-Ri (1.127 mg; S.E.M.: ±0.020 mg) groups, respectively (*Figure 1A*). Similarly, male *TubG4ts*>*CTPS*-Ri flies (0.745 mg; S.E.M.:±0.017 mg) weighed 9.8%, 9.1%, and 7.7% less than those in the *TubG4ts*>+ (0.826 mg; S.E.M.: ±0.017 mg), *CTPS*-Ri/+ (0.82 mg; S.E.M.: ±0.009 mg) and *TubG4 ts*>Con-Ri (0.807 mg; S.E.M.: ±0.065 mg) groups, respectively (*Figure 1A*).

Next, we knocked down *CTPS* specifically in the central nervous system using *ElavG4* (*Elav GAL4*). In contrast to the global knockdown flies, the body weights of female and male *ElavG4*>*CTPS*-Ri flies did not differ significantly from those of the corresponding *ElavG4*>+, *CTPS*-Ri/+, and *ElavG4*>Con-Ri control flies (*Figure 1B*).

Finally, we used the *CgG4* (*CgGAL4*) driver to knock down *CTPS* in the fat body specifically. The results showed that the body weights of female and male *CgG4*>*CTPS*-Ri flies were significantly less than those of the *CgG4*>+, *CTPS*-Ri/+, and *CgG4*>Con-Ri control flies (*Figure 1C*). Specifically, the body weight of the *CgG4*>*CTPS*-Ri female flies (0.859 mg; S.E.M.: ±0.148 mg) was 29%, 26.7%, and 22.4% less than those of the *CgG4*>+ (1.21 mg; S.E.M.: ±0.026 mg), *CTPS*-Ri/+ (1.172 mg; S.E.M.: ±0.079 mg), and *CgG4*>Con-Ri (1.108 mg; S.E.M.:±0.134 mg) flies. Similarly, the body weight of the *CgG4*>*CTPS*-Ri male flies (0.740 mg; S.E.M.: ±0.027 mg) was also significantly reduced: 12.9%, 9.8%, and 10.8% less than those of the *CgG4*>+ (0.85 mg; S.E.M.: ±0.027 mg), *CTPS*-Ri/+ (0.82 mg; S.E.M.: ±0.009 mg), and *CgG4*>Con-Ri (0.83 mg; S.E.M.: ±0.002 mg) lines, respectively (*Figure 1C*).

The differential impact of CTPS knockdown on body weight loss in various tissues may be attributed to differences in the strength and pattern of the GAL4 driver used. We then utilized quantitative RT-PCR (qRT-PCR) to determine the efficiency of *CTPS* knockdown in different tissues. The lower efficiency of *CTPS* knockdown in the entire body (33–44%) compared to the fat body (62–64%) (*Figure 1D and F* and *Figure 1—figure supplement 1A,C*) may account for the weakened body weight loss observed in *TubG4ts*>*CTPS*-Ri flies (*Figure 1A and C*). Conversely, even though CTPS knockdown was more pronounced in pan neuron cells (75–82%) than in the fat body (*Figure 1E and F* and *Figure 1—figure supplement 1B,C*), this knockdown did not reduce body weight, indicating that CTPS in the fat body is necessary to facilitate weight gain.

### CTPS is required for starvation resistance

In laboratory experiments, it is commonly observed that an increase in body weight reflects an increase in energy reserves, particularly in lipid stores, which is an adaptive response to starvation (*Rion and Kawecki, 2007*). Our investigation aimed to determine the impact of CTPS on starvation response, and we found that female and male *TubG4ts*>*CTPS*-Ri flies had considerably shorter survival durations when starved. Specifically, when *TubG4ts*> *CTPS*-Ri flies were compared to *TubG4ts*>+, *CTPS*-Ri/+, and *TubG4ts*>Con-Ri flies, female median survival rates declined by 30.2%, 25%, and 40%, respectively, whereas male median survival rates fell by 50%, 20%, and 20%, respectively (*Figure 1G*).

No significant difference in starvation resistance was observed between the *ElavG4*>*CTPS*-Ri flies and the *CTPS*-Ri/+flies (*Figure 1H*). However, *ElavG4*>*CTPS*-Ri flies displayed decreased survival deficits in starvation conditions when compared to *ElavG4* >+ and *ElavG4*>Con-Ri flies. Specifically, female median survival rates declined by 20% and 11%, respectively, whereas male median survival rates decreased by 26.7% and 26.7%, respectively (*Figure 1H*).

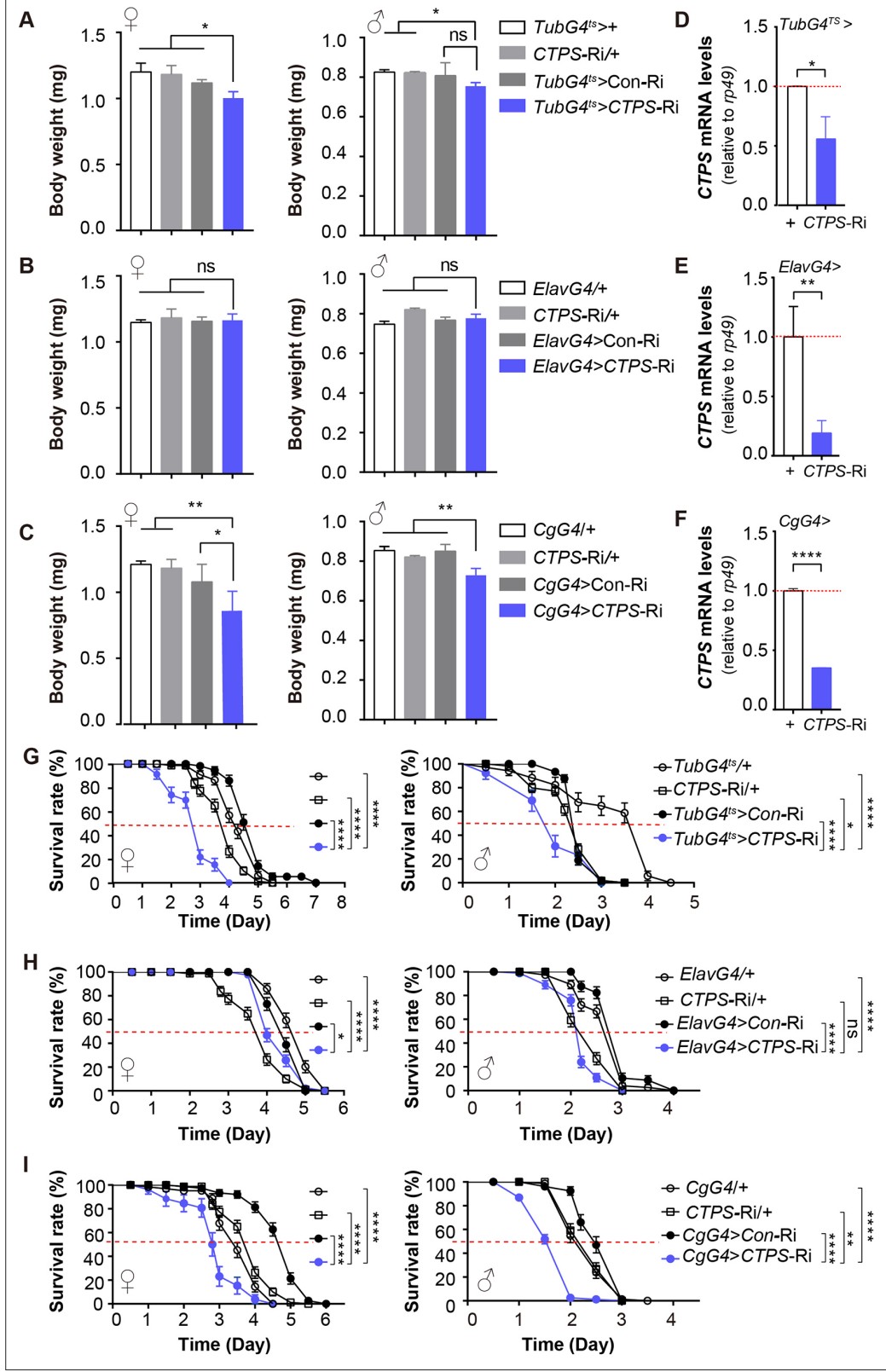

**Figure 1.** CTPS knockdown in the *Drosophila* fat body leads to body weight loss. (**A–C**) Body weights of 5-day-old adult flies from the indicated genotypes (30 flies/group, 5–6 groups/genotype, 2–3 biological replicates). *TubG4ᵗˢ>CTPS*-Ri versus *TubG4ᵗˢ>+*, *CTPS*-Ri/+ or *TubG4ᵗˢ>Con*-Ri (**A**), *ElavG4 >CTPS*-Ri versus *ElavG4>+*, *CTPS*-Ri/+ or *ElavG4>CoRi* (**B**), and *CgG4>CTPS*-Ri versus *CgG4>+*, *CTPS*-Ri/+ or *CgG4>Con*-Ri (**C**). All values

*Figure 1 continued on next page*

*Figure 1 continued*

are the means ± standard error of the mean (S.E.M.). ns, not significant, * P<0.05, ** P<0.01, and *** P<0.001 in one-way ANOVA with a Tukey *post hoc* test. (**D–F**) Quantitative RT-PCR analysis of the mRNA abundance of *CTPS* from whole-body (**D, F**) or head (**E**) lysates of adult flies in indicative lines (10 flies/group, 3 groups/genotype, 3 biological replicates). *TubG4ᵗˢ>CTPS* Ri versus *TubG4ᵗˢ>+* (**D**); *ElavG4>CTPS*-Ri versus *ElavG4>+* (**E**); *CgG4>CTPS*-Ri versus *CgG4>+* (**F**). Relative value are normalized with the control line. All values are the means ± S.E.M. * P<0.05, ** P<0.01, and **** P<0.0001 by Student's t test. (**G–I**) Survival curves for starved female and male adult flies from the indicated genotypes (5 days of age; 30 flies/group, 5 groups/genotype, 3 biological replicates). Graphs represent percent survival as the calculated mean survival rate of each group. ns, no significance, *P<0.05, **P<0.01, and ****P<0.0001 by log-rank test.

The online version of this article includes the following figure supplement(s) for figure 1:

**Figure supplement 1.** Quantitative RT-PCR analysis.

**Figure supplement 2.** Inhibition of *CTPS* in adipocytes reduced body weight and resistance to starvation.

**Figure supplement 3.** TAG level of male adults upon food deprivation.

Moreover, female and male *CgG4>CTPS*-Ri flies showed reduced survival durations in starvation conditions when compared to *CgG4>+*, *CTPS*-Ri/+, and *CgG4>Con*-Ri flies. Specifically, female median survival rates declined by 17.1%, 27.5%, and 42%, respectively, whereas male median survival rates decreased by 25%, 25%, and 25%, respectively (*Figure 1I*).

To address the possibility of leakage in expression, we employed another fat body driver line, *pplG4*. After food restriction, the *pplG4>CTPS*-Ri flies exhibited declines in body weight and survival comparable to those seen for the *CgG4>CTPS*-Ri flies (*Figure 1—figure supplement 2A and B*).

To investigate whether the sensitivity of *CgG4>CTPS*-Ri flies to starvation was due to insufficient lipid storage, we measured triglyceride (TAG) levels in male adults. Under adequate nutritional conditions, the TAG content of *CgG4>CTPS*-Ri flies was reduced by 74.1%, 83.5%, and 62.5% when compared to *CgG4>+*, *CTPS*-Ri/+, and *CgG4>Con*-Ri flies, respectively (*Figure 1—figure supplement 2A*). The TAG levels in flies declined gradually when they were starved, and in *CgG4>CTPS*-Ri flies, TAG was almost completely depleted after 24 hour food deprivation (*Figure 1—figure supplement 3A*). These results explain the lower median survival rate of *CgG4>CTPS*-Ri flies compared to the control lines (*Figure 1I*), and suggest that the lack of starvation resistance observed when CTPS is deficient may be due in part to inadequate TAG storage.

## CTPS in the fat body is crucial for body weight maintenance in larvae

To investigate the effect of CTPS on larval body weight and fat mass, we examined wandering stage larvae using body weight and the floating assay. We observed that *CTPS* knockdown significantly decreased larval body weight when compared to that measured in *CgG4>+*, *CTPS*-Ri/+, and *CgG4>Con*-Ri control lines (*Figure 2A*), while having no effect on larval body size (*Figure 2—figure supplement 1A*). Furthermore, we used a rapid floating assay to compare fat content in *Drosophila* larvae. This assay is based on the principle that individuals with a higher fat content float better than lean individuals in a solution of fixed density (*Liu et al., 2012*; *Reis et al., 2010*). Our results revealed that 80% of *CgG4>CTPS*-Ri larvae sank to the bottom of the vial, whereas 80%–90% of the control larvae floated on top of the approximately 12% sugar solution (*Figure 2B*).

To ensure that our results were not due to off-target effects, we also utilized another *CTPS* RNAi line, *CTPS*-Riᵀᴿⁱᴾ·ᴶᶠ⁰²²¹⁴. We observed a significant reduction in both body weight (*Figure 2C*) and floating rate (*Figure 2D*) in either *pplG4>CTPS*-Riᵀᴿⁱᴾ·ᴴᴹ⁰⁴⁰⁶² or *pplG4>CTPS*-Riᵀᴿⁱᴾ·ᴶᶠ⁰²²¹⁴ larvae when compared to *pplG4>+*larvae. We evaluated the knockdown efficiency of *CTPS* in the two lines using qRT-PCR (*Figure 2—figure supplement 2A*). In addition, no developmental delay was observed in any of the larval stages in the *CgG4>CTPS*-Ri line.

## HFD promotes CTPS expression in the fat body

To investigate how CTPS cytoophidia change in adipocytes in response to HFD, we utilized mCherry and V5-tagged *CTPS* knock-in larvae (*CTPS*-mCh) (*Liu et al., 2022*), which were fed a HFD containing 30% coconut oil to stimulate lipogenesis in the fat body. We first evaluated the *CTPS* expression level in fat bodies from HFD-fed or regular diet (RD)-fed larvae using qRT-PCR. Our results demonstrated that *CTPS* expression in the fat body is upregulated by 120% under HFD feeding compared to RD

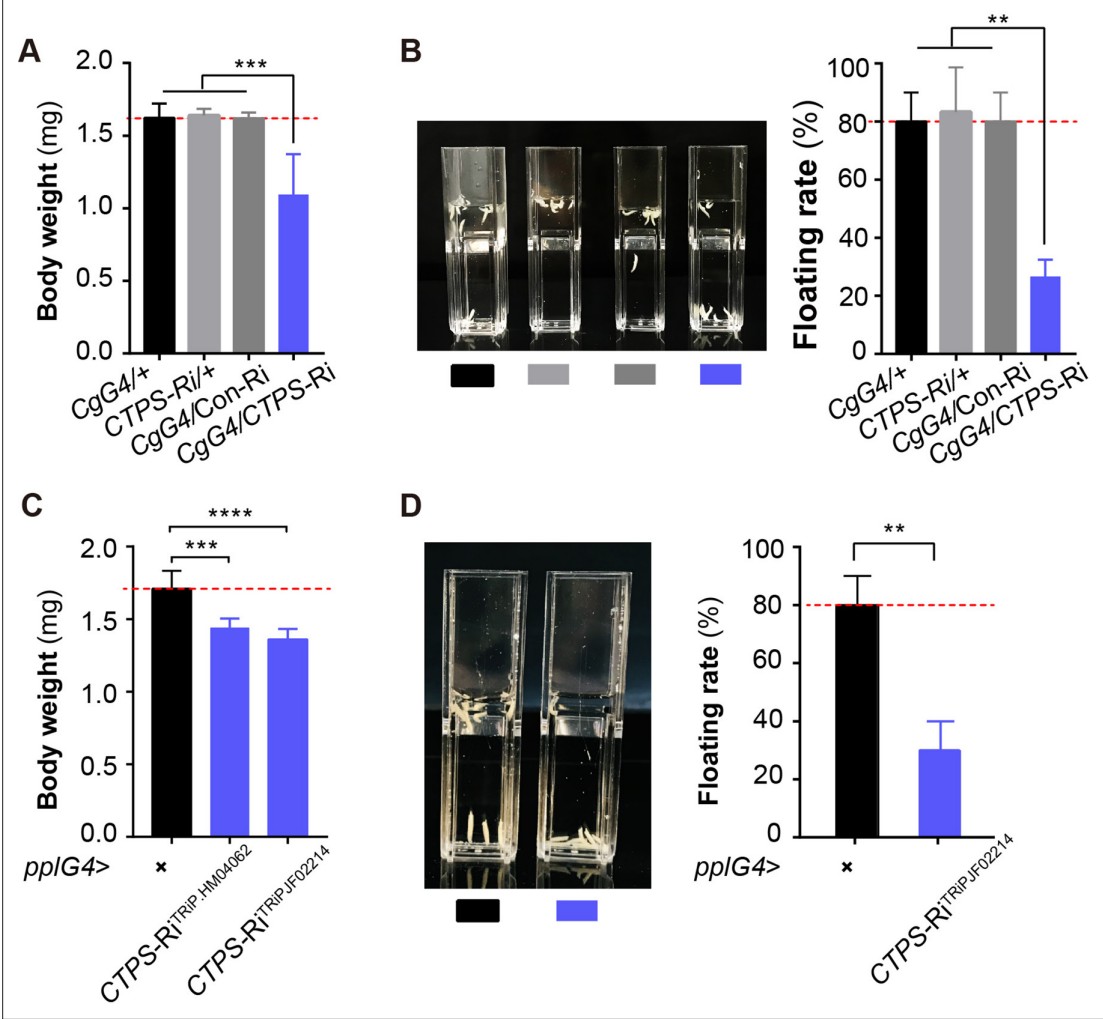

**Figure 2.** Adipocyte-specific knockdown of *CTPS* decreases larval body weight. (**A**) The 3rd instar wandering larval body weight of the indicated lines (10 larvae/group, 5–6 groups/genotypes, 3 biological replicates). *CgG4>CTPS*-Ri is compared with *CgG4>*+, *CTPS*-Ri/+ or *CgG4>*Con-Ri controls. (**B**) Representative photograph of the floating assay (10 larvae/group, 3 groups/genotype, 3 biological replicates) and quantification of floatation scores (% floating larvae, right panel). *CgG4>CTPS*-Ri is compared with *CgG4>*+, *CTPS*-Ri/+ or *CgG4>*Con-Ri control lines. (**C**) 3rd instar wandering larval body weight of *pplG4>CTPS*-Ri and *pplG4>*+ lines (10 larvae/group, 3 groups/genotype, 3 biological replicates). (**D**) Representative photograph of the floating assay (10 larvae/group, 3 groups/genotype, 3 biological replicates), and the quantification of floatation scores (% floating larvae, right panel). *pplG4>CTPS*-Ri and *pplG4>*+ lines are compared. Data are shown as means ± S.E.M. ** $P<0.01$, *** $P<0.001$, and **** $P<0.001$ by Student's t test.

The online version of this article includes the following figure supplement(s) for figure 2:

**Figure supplement 1.** Larval size comparison.

**Figure supplement 2.** Quantitative RT-PCR analysis.

conditions (*Figure 3A*). In addition, we observed that the CTPS cytoophidia in the fat body of HFD-fed *CTPS*-mCh larvae were elongated by up to 60% (*Figure 3B–D*) when compared to those in RD-fed larvae, while cytoophidia numbers were also increased modestly (*Figure 3E*). These findings suggest that elongated cytoophidia and elevated CTPS in the fat body may facilitate metabolic adaptation in response to a HFD.

## Fat-body-specific knockdown of CTPS alleviates HFD-induced obesity

To investigate the impact of CTPS on lipogenesis and metabolic adaption in the fat body, we used *CgG4* in combination with UAS-eGFP to indicate fat mass. To reduce the potential variability in larval developmental timing during the experiment, we restricted the collection of eggs to 4 hours. We cultured them until they reached a specific developmental stage. Specifically, we harvested the larvae

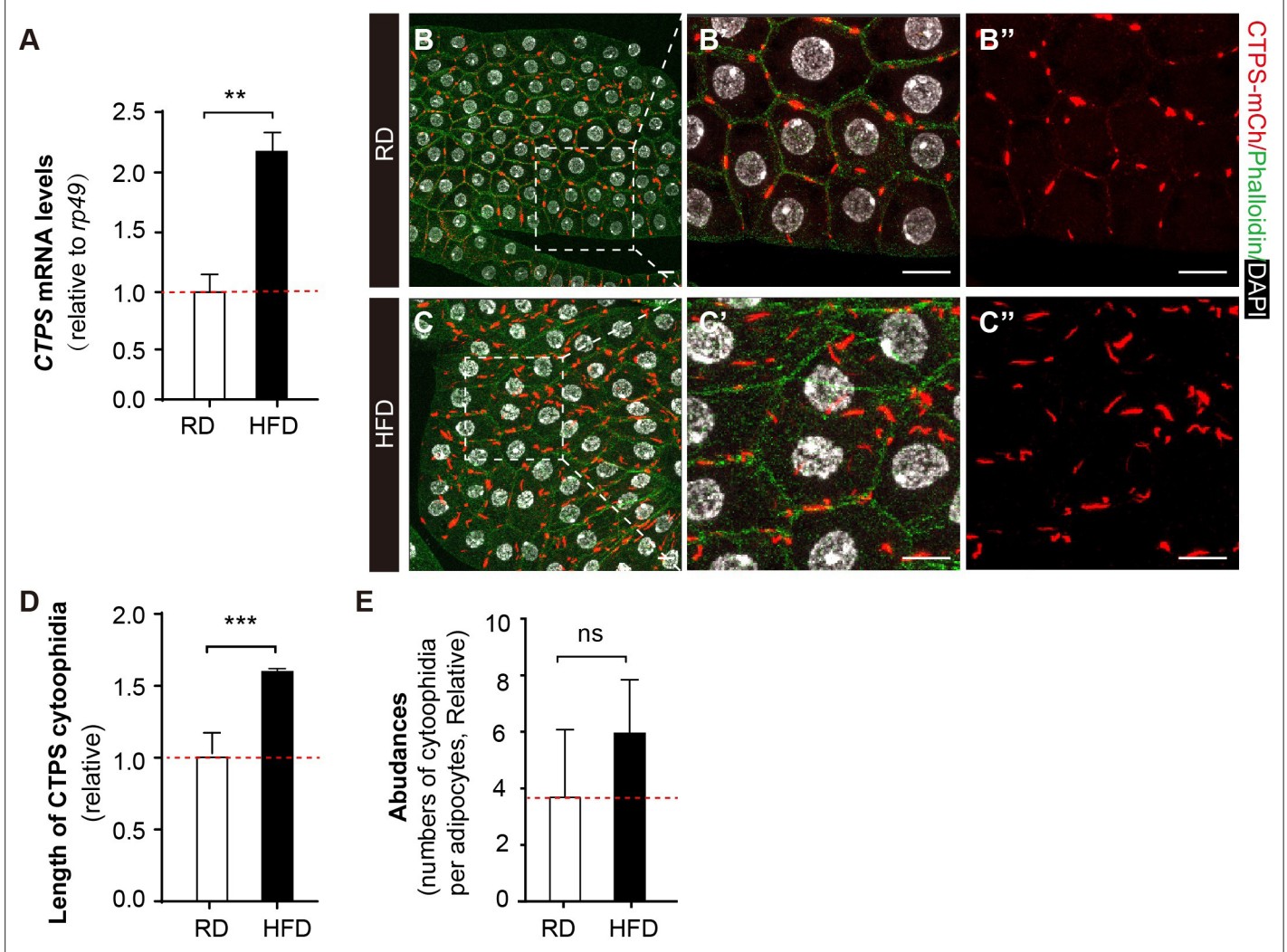

**Figure 3.** HFD promotes CTPS expression in the fat body. (**A**) Quantitative RT-PCR analysis of the abundance of *CTPS* mRNA in fat body lysates of 76~80 hour after egg laying (AEL) larvae under RD and HFD conditions. The relative value is normalized with larvae under RD feeding (30 larvae/group; 3 groups/genotype; 3 biological replicates). (**B–C**) Representative confocal images of fat bodies from the 80 hour AEL larvae show that CTPS cytoophidia showed increased elongation upon HFD feeding (C, C', and C") when compared to those in RD-fed larvae (B, B', and B") (20 images/genotype; 3 biological replicates). The area within the white square is magnified in the right panel (B', B", C', and C"). Cell plasma membranes are stained with phalloidin (green). Nuclei are stained with DAPI (white). Scale bar, 20 µm. (**D**) Quantification of the length of the cytoophidia shown in (**B, C**). The relative value is normalized with larvae under RD feeding (20 images/genotype; 3 biological replicates). (**E**) Quantification of the numbers of cytoophidia per adipocyte shown in (**B, C**). The relative value is normalized with larvae under RD feeding (20 images/genotype; 3 biological replicates). All values are the means ± S.E.M. ns, no significance, ** p<0.01, and *** *P*<0.001 by Student's *t*-test.

at 76 hours after egg laying (AEL). After feeding a HFD, we observed a significant increase in eGFP intensity in wild-type larvae, indicating that HFD induces robust lipogenesis and provides a suitable model for studying the impact of CTPS on fat metabolism (*Figure 4A and A"*). However, *CgG4*, eGFP>*CTPS*-Ri larvae did not show a significant increase in eGFP intensity following HFD feeding when compared to the control lines (*Figure 4A and A"*). We then examined eGFP transcript levels in the whole body or in the fat body using qRT-PCR. Although there was no apparent change in the fat body, eGFP transcript levels were dramatically decreased in the enitre body of *CgG4*, eGFP>*CTPS*-Ri larvae (*Figure 4—figure supplement 1A*). Dissecting out the fat body revealed that the total amount of fat body in HFD-fed *CgG4*, eGFP>*CTPS*-Ri larvae was significantly smaller than that in *CgG4*, eGFP>+ larvae (*Figure 4A'*), which explained the lower eGFP intensity and reduced eGFP transcript levels in the whole body of *CgG4*, eGFP>*CTPS*-Ri larvae (*Figure 4A and A"*). *CgG4* >+ larvae fed HFD gained significantly more body weight (429%, 1.550 mg; S.E.M.: ±0.021 mg) than those on a

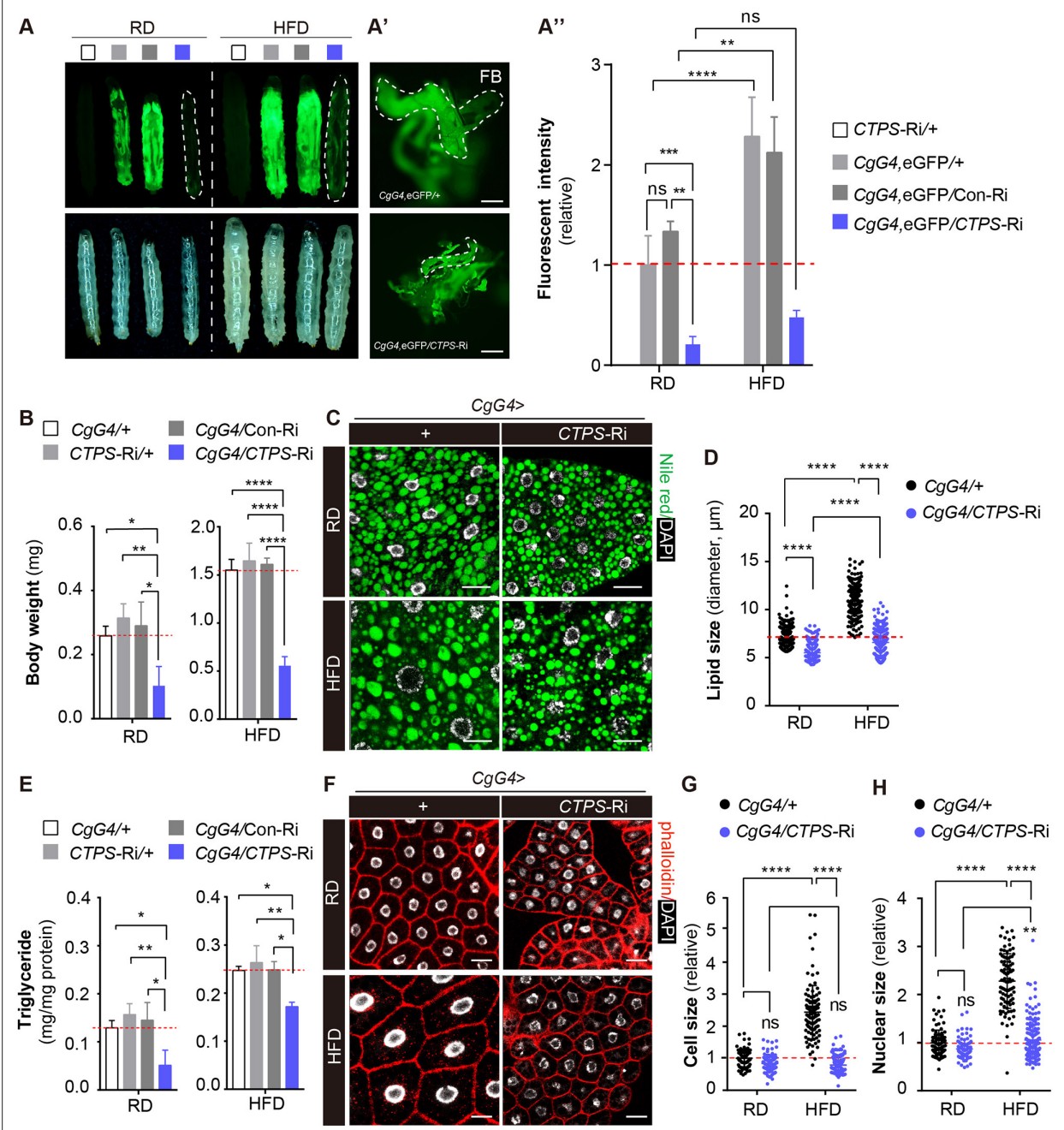

**Figure 4.** Knockdown of *CTPS* in adipocytes alleviates HFD-induced obesity. (**A**) 76~80 hour AEL larvae expressing eGFP (green) with *CgG4* driving *CTPS* knockdown in the fat body and the wild-type control were fed with RD or HFD (eGFP fluorescent image top, bright-field image bottom). Dashed lines denote the extent of the larval bodies. (**A'**) Photographs of newly dissected larval fat bodies (FB) (green, eGFP-labelling). Scale bars, 500 μm. (**A''**) Quantification of eGFP intensity from (**A**). The values are normalized to the control line *CgG4*, eGFP>+ (5 images/genotype; 3 biological replicates). (**B**) The body weight of the 76~80 hour AEL larvae under RD and HFD conditions: *CgG4>CTPS*-Ri larvae are compared with *CgG4>+*, *CTPS*-Ri/+, and *CgG4>*Con-Ri larvae (10–30 larvae/group; 5–6 groups/genotype; 3 biological replicates). (**C**) Lipid droplets from 76~80 hour AEL larvae fed RD and HFD or HFD were analyzed by confocal microscopy. Lipid droplets were stained with Nile red (green), and nuclei were stained with DAPI (white). Scale bars, 20 μm. (**D**) Quantitative analyses of lipid droplet size from (**F**) (10 images/genotype; 3 biological replicates). (**E**) TAG level of 80 hour AEL larvae from *CgG4>CTPS*-Ri and *CgG4* >+ lines under RD and HFD conditions. TAG level is normalized to total protein level (10 larvae/group; 3–4 groups/genotype; 2 biological replicates). (**F**) Confocal images of fat bodies from 76~80 hour AEL larvae under RD and HFD conditions. Phalloidin (red) is used to reveal cell outline and DAPI (white) is used to reveal the nuclei in fat bodies. Scale bars, 20 μm. (**G**) Quantification of cell size from (**F**) (10 images/genotype; 3 biological replicates). (**H**) Quantification of nuclear size from (**F**) (10 images/genotype; 3 biological replicates). Data are shown as mean ±S.E.M. ns, no significance, * $P<0.05$, ** $P<0.01$, and **** $P<0.0001$ by one-way ANOVA or two-way ANOVA with a Tukey *post hoc* test.

*Figure 4 continued on next page*

*Figure 4 continued*

The online version of this article includes the following figure supplement(s) for figure 4:

**Figure supplement 1.** Quantitative RT-PCR analysis.

**Figure supplement 2.** CTPS is required for adipocyte growth.

**Figure supplement 3.** The effects of CTPS on phospholipid composition and the expression of nucleotide diphosphate kinases.

RD (0.293 mg; S.E.M.: ±0.011 mg) (*Figure 4B*). However, HFD-fed *CgG4>CTPS*-Ri larvae (0.549 mg; S.E.M.: ±0.096 mg) exhibited a 64.6% decrease in body weight when compared to HFD-fed *CgG4>+* larvae (*Figure 4B*). *CTPS* knockdown resulted in a 73% reduction in body weight gain when compared to *CgG4>+* larvae when both lines were fed HFD (*Figure 4B*). The body weights of *CTPS*-Ri/+ and *CgG4>*Con-Ri control lines were similar to those of *CgG4 >+* larvae under both HFD and RD conditions.

We utilized Nile red staining to visualize the lipid droplets in the larval fat body and quantified the TAG level in adipocytes. *CgG4>+* larvae displayed larger lipid droplets in adipocytes after HFD feeding, whereas *CgG4>CTPS*-Ri showed significantly smaller lipid droplets in adipocytes compared to the control groups, regardless of the diet (*Figure 4C and D*). In RD conditions, the TAG content of *CgG4>CTPS*-Ri larvae was 50%, 58.8%, and 55.4% lower than that of the *CgG4>+*, *CTPS*-Ri/+, or *CgG4>CTPS*-Ri control group, respectively (*Figure 4E*). Similarly, under HFD conditions, the TAG content in *CgG4>CTPS*-Ri larvae was 30.1%, 35%, and 31% lower than in the *CgG4>+*, *CTPS*-Ri/+, or *CgG4>CTPS*-Ri control groups, respectively (*Figure 4E*). In addition, we observed a remarkable reduction in HFD-induced adipocyte expansion in *CgG4>CTPS*-Ri larvae, as evidenced by smaller cell and nuclear sizes than those in the control larvae (*Figure 4F–H*). To eliminate any potential systemic feedback effects, we performed a clonal analysis of the fat body. By crossing the *yw, hs-flp; act>CD2>G4*, UAS-GFP line with the *CTPS*-Ri line and inducing *CTPS* knockdown by heat shock, we generated CTPS-deficient fat body cell clones. Remarkably, the clones expressing CTPS RNAi were considerably smaller than their control clones (*Figure 4—figure supplement 2A, B and C*), indicating that CTPS is required for adipocytes to sustain growth cell-autonomously.

We investigated whether CTPS deficiency affects phospholipid synthesis in the fat body. We profiled phospholipid levels in the fat body of larvae expressing *CTPS*-Ri. We found that the levels of the major phospholipids, including phosphatidylethanolamine (PE), phosphatidylcholine (PC), lyso-phosphatidylethanolamine (LPE), and lysophosphatidylcholine (LPC), were slightly reduced (although not significantly) when normalized to protein concentration (*Figure 4—figure supplement 3A*). A reduction in phospholipid biosynthesis resulting from CTPS deficiency could lead to smaller lipid droplets in adipocytes. We also examined the expression levels of genes such as *nmydn-D6*, *nmydn-D7*, and *CG15547*, which encode the nucleotide diphosphate kinases that catalyze the conversion of CDP to CTP. We found that the expression of *nmydn-D7* was increased in fat bodies expressing *CTPS*-Ri (*Figure 4—figure supplement 3B*), indicating that fat bodies may upregulate nucleotide diphosphate kinase expression to compensate for CTP production when de novo CTP synthesis is hindered. Taken together, our findings indicate that CTPS is essential for adipocyte expansion and lipogenesis in response to HFD consumption.

## Fat-body-specific knockdown of CTPS reduces lipogenic gene expression

To better understand how CTPS affects adipocyte function and lipid homeostasis, we conducted a genome-wide RNA sequencing (RNA-Seq) analysis of larval adipocytes with CTPS knockdown (*CgG4>CTPS*-Ri) and wild-type controls (*CgG4>+*). Our analysis identified 273 differentially expressed genes, with 204 genes (74.7%) upregulated and 69 genes (25.3%) downregulated in CTPS-deficient adipocytes compared to controls (≥2.0-fold change, Student's *t*-test, $P<0.05$) (*Figure 5—figure supplement 1A*). Kyoto Encyclopedia of Genes and Genomes (KEGG) analysis indicated that the differentially regulated genes were significantly enriched in lipid metabolism, carbohydrate metabolism, and amino acid metabolism (*Figure 5—figure supplement 1B*). Notably, we observed the downregulation of genes encoding lipogenic enzymes such as acetyl-CoA carboxylase (ACC) and fatty acid synthase 1 (FASN1), and of several other genes that are involved in lipid metabolism (*Figure 5A*).

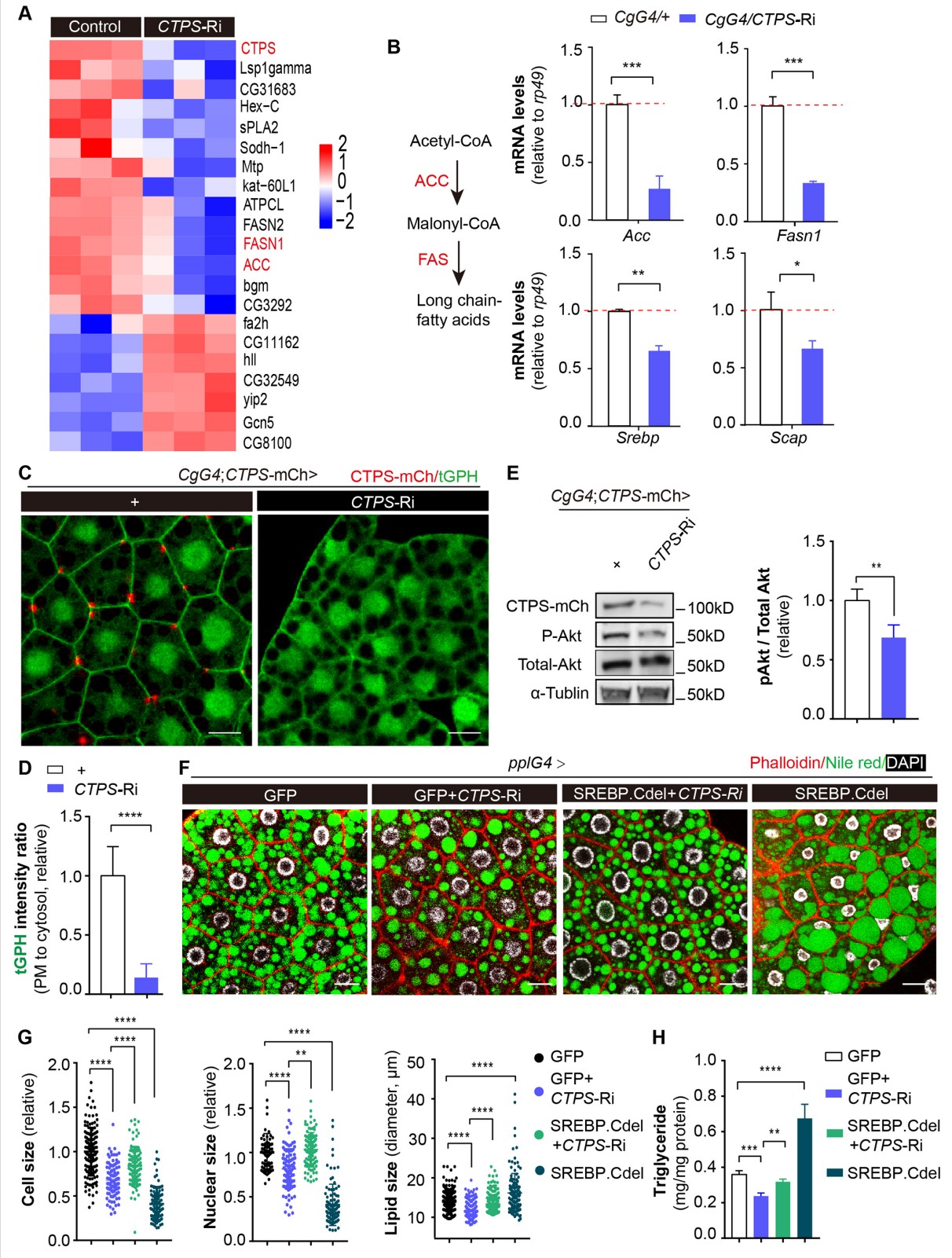

**Figure 5.** Fat-body-specific knockdown of CTPS reduces lipogenic gene expression. (**A**) The fat bodies of 2nd instar larvae from *CgG4>CTPS*-Ri and *CgG4>+* larvae were analyzed by RNA-seq analysis. A heat map of relative gene expression, obtained using RNA-seq data, is depicted for transcripts encoding central enzymes in lipid metabolism from control (left) and *CTPS* knockdown (right) larvae. (**B**) Quantitative RT-PCR analysis of the mRNA abundance of *Acc*, *Fasn1*, *Srebp*, and *Scap* in fat body lysates from 76~80 hour AEL *CgG4>CTPS*-Ri and *CgG4>+* larvae (30 larvae/group; 5–6 groups/

*Figure 5 continued on next page*

*Figure 5 continued*

genotype; 3 biological replicates). The long-chain fatty acid synthesis pathway is shown in the left panel. (**C**) Representative confocal images of PI3K activation in the fat bodies of 76~80 hour AEL larvae. The membrane location of tGPH (green) shows the activity of PI3K. Scale bars, 10 μm. (**D**) tGPH intensity ratio of the cell membrane to the cytosol from (**C**). The value is normalized to the control (10 images/genotype; 3 biological replicates). (**E**) Western blot analysis of phosphorylated Akt from fat body lysates. Anti-mCh, anti-phosphorylated-Akt, and anti-total-Akt antibodies were used for the immunoblotting analysis. Alpha-tubulin was used as an internal control. The P-Akt to total-Akt ratio is shown (right panel). The value is normalized to the *CgG4;CTPS*-mCh>+ control line (3 biological replicates). (**F**) Representative confocal images of 96~100 hour AEL larval fat bodies. Fat bodies are stained with phalloidin (red) to reveal the cell outline, Nile red (green) to reveal lipid droplets, and DAPI (white) to reveal nuclei. Scale bars, 30 μm. (**G**) Quantification of cell size, nuclear size and lipid droplet size from (**F**). Cell and nuclear sizes are normalized to the *pplG4*>GFP control line (10 images/genotype; 3 biological replicates). (**H**) TAG concentration in 96~100 hour AEL larvae under HFD conditions. TAG level is normalized to total protein level (6 larvae/group; 5–6 groups/genotype; 2 biological replicates). All data are shown as mean ± S.E.M. * $P<0.05$, ** $P<0.01$, *** $P<0.001$, and **** $P<0.0001$ by Student's *t*-test or one-way ANOVA with a Tukey *post hoc* test.

The online version of this article includes the following source data and figure supplement(s) for figure 5:

**Source data 1.** List of the differentially expressed genes in the heatmap.

**Source data 2.** Uncropped gel of phosphorylated Akt from fat body lysates.

**Figure supplement 1.** KEGG functional classification of the genes that are affected by CTPS knockdown.

**Figure supplement 1—source data 1.** List of baseMean values for the *CgG4*>*CTPS*-Ri line relative to the *CgG4*>+ control line.

**Figure supplement 1—source data 2.** Numbers of genes belonging to distinct functional groups that are up- or downregulated in the *CgG4*>*CTPS*-Ri line relative to the *CgG4*>+ control line.

**Figure supplement 2.** Quantitative RT-PCR analysis.

**Figure supplement 3.** Quantitative RT-PCR analysis.

To validate our RNA-Seq results, we used qRT-PCR to measure the expression levels of genes in wild-type and CTPS-deficient larval adipocytes. Our results showed that *CTPS* knockdown significantly decreased the expression levels of *Acc* and *Fasn1* by 54–86.1% and 59–70.3%, respectively (**Figure 5B**, **Figure 5—figure supplement 2A**, B). Lipogenesis is regulated by various factors, including sterol regulatory element binding protein (SREBP), a highly conserved membrane-bound transcription factor that plays a critical role in the transcriptional regulation of lipogenic enzymes (**Seegmiller et al., 2002**). SREBP forms a complex with SREBP cleavage-activating protein (SCAP) in the endoplasmic reticulum (ER). The SREBP–SCAP complex then moves into the Golgi system for processing and cleaved SREBP translocates into the nucleus to increase the transcription of several genes that are involved in fatty-acid synthesis, such as *Acc* and *Fasn1*. We investigated whether CTPS knockdown also affects the transcriptional levels of *Srebp* and *Scap*. Our qRT-PCR results showed that the expression levels of *Srebp* and *Scap* were reduced by 23–34.4% and 30–33.3%, respectively, in response to CTPS knockdown (**Figure 5B**, **Figure 5—figure supplement 2A,B**). These findings suggest that CTPS plays a role in regulating adipocyte lipogenesis by modulating the expression of lipogenic enzymes.

## Fat-body-specific knockdown of CTPS suppresses PI3K-Akt signaling

The phosphatidylinositol 3'-kinase (PI3K) phosphorylates inositol lipids in membranes, facilitating intracellular signal transmission (**Engelman et al., 2006**). PI3K modulates SREBP activity through the Akt signal, increasing the growth of fat bodies and driving various aspects of cell metabolism, such as lipid storage (**Krycer et al., 2010**; **Luu et al., 2012**; **Yecies et al., 2011**). CTPS affects the distribution of integrins at the adipocyte membrane (**Liu et al., 2022**). A pathway linking Integrin signaling to Akt activation via PI3K has been identified (**Zeller et al., 2010**), prompting us to investigate whether CTPS modulates PI3K–Akt signaling in the fat body. To track the PI3K activity of larval adipocytes, we utilized tGPH as a cytological marker (**Britton et al., 2002**). When comparing the ratio of tGPH in the cell membrane to that in the cytosol, we observed a significant reduction in the cell membrane-associated tGPH signal in the adipocytes of *CgG4*>*CTPS*-Ri (**Figure 5C and D**). PI3K controls the membrane localization of tGPH. We then employed qRT-PCR to measure the expression level of four PI3K subunits in the fat body. Our results demonstrated that when *CTPS* was knocked down, the expression level of *Pi3K* was not significantly reduced (**Figure 5—figure supplement 3A**), suggesting that CTPS depletion reduces PI3K activity rather than *Pi3K* expression level, which diminishes tGPH localization to the cell membrane.

The activity of Akt is directly targeted by the PI3K signal. Therefore, we hypothesized that *CTPS* deficiency may result in reduced Akt activity as a result of decreased PI3K activity. To test this hypothesis, we assessed Akt phosphorylation by detecting the phosphorylation of fly Akt at Ser505, a site conserved with Ser473 of murine Akt1. We observed a significant decrease in the level of phosphorylated Akt in the fat body of *CgG4>CTPS*-Ri larvae (*Figure 5E*). Furthermore, we investigated whether activating SREBP could restore the impaired fat metabolism that results from CTPS depletion under HFD feeding. Intriguingly, we found that overexpressing the truncated form of SREBP.Cdel, constitutively activated and nuclear localized, partially rescued the reduction in adipocyte size and the smaller lipid droplet size caused by *CTPS* deficiency (*Figure 5F and G*). Moreover, we observed that the TAG concentration was partially restored in larvae overexpressing SREBP.Cdel in the absence of CTPS, but did not reach the same level as that in the wild-type control (*Figure 5H*). It is important to highlight, however, that the overexpression of SREBP.Cdel alone led to a noteworthy reduction in both adipocyte and nuclear size, even though there was a marked increase in TAG accumulation. This observation could provide a possible reason to explain why the overexpression of activated SREBP did not fully rescue all of the defects caused by CTPS knockdown. It suggests that although overexpression of SREBP can stimulate lipogenesis, it may not increase cell size. The precise expression of SREBP may be crucial for regulating cell size. Collectively, our results suggest that CTPS is crucial for the control of lipogenesis, potentially by preserving the activation of PI3K-Akt-SREBP signaling.

## Disrupting the filament-forming property of CTPS alleviates HFD-induced obesity

We were intrigued by the potential function of CTPS cytoophidia in adipocytes during HFD feeding. Our research revealed that H355 in the domain of glutamine amidotransferase is critical for *Drosophila* cytoophidium formation (*Zhou et al., 2019*; *Zhou et al., 2021a*). We generated transgenic fly lines expressing mCherry-HA tagged wild-type CTPS or CTPS with a point mutation (H355A). Using the *CgG4* driver, we specifically overexpressed wild-type (CTPS^WT-OE) or H355A mutant CTPS (CTPS^MU-OE) in adipocytes. The expression levels of CTPS in both the CTPS^WT-OE and the CTPS^MU-OE lines were comparable (*Figure 6—figure supplement 1A*). CTPS^MU-OE significantly reduced HFD-induced increases in body weight (*Figure 6A and B*) and TAG accumulation (*Figure 6C*) compared to the control lines. Specifically, CTPS^MU-OE (1.293 mg; S.E.M.: ±0.146 mg) resulted in body weight losses of 18.8%, 16.7%, and 18.8%, respectively, when compared to *CgG4>+* (1.59 mg; S.E.M.: ±0.116 mg), *CgG4>*GFP (1.552 mg; S.E.M.: ±0.225 mg), and CTPS^MU/+ (1.592 mg; S.E.M.: ±0.145 mg) control lines (*Figure 6B*). In addition, the TAG content in CTPS^MU-OE was reduced by 19.8–24.8% compared to the control lines (*Figure 6C*). Conversely, there were no significant differences in body weight or TAG content between CTPS^WT-OE (1.558 mg; S.E.M.:±0.188 mg) and the *CgG4>+*, *CgG4>*GFP, and CTPS^WT/+ (1.593 mg; S.E.M.: ±0.139 mg) control lines (*Figure 6B and C*), indicating that cytoophidia are required but not sufficient for body weight gain in response to HFD consumption.

The confocal images of the fat body stained with phalloidin and BODIPY 493/503 showed that the control group and CTPS^WT-OE enhanced adipocyte expansion and lipid droplet accumulation in response to HFD. However, CTPS^MU-OE larvae exhibited a significant decrease in cell size and lipid size (*Figure 6D and E*). The overexpression of CTPS^MU prevented the formation of cytoophidia in adipocytes (*Figure 6D and F*), which is in line with our previous research (*Liu et al., 2022*). In addition, CTPS^MU overexpression reduced the membrane location of tGPH in adipocytes by 55% (*Figure 6F and G*), indicating a considerable downregulation of PI3K activity in the absence of cytoophidia. The phosphorylation of Akt was also diminished in CTPS^MU-OE adipocytes (*Figure 6H*). Moreover, when compared with *CgG4>+*, there was a significant decrease of 40.2–65.3% and 30.3–40.1% in the expression levels of *Acc* and *Fasn1*, respectively, in CTPS^MU-OE adipocytes (*Figure 6I*, *Figure 6—figure supplement 2A*). However, no significant changes were found in *Fasn1* mRNA levels in CTPS^WT-OE (*Figure 6I*, *Figure 6—figure supplement 2A*). These results suggest that the loss of CTPS cytoophidia impedes the PI3K-Akt signaling pathway and suppresses lipogenic genes, leading to a reduction in adipocyte expansion and inhibition of lipogenesis induction in response to HFD consumption.

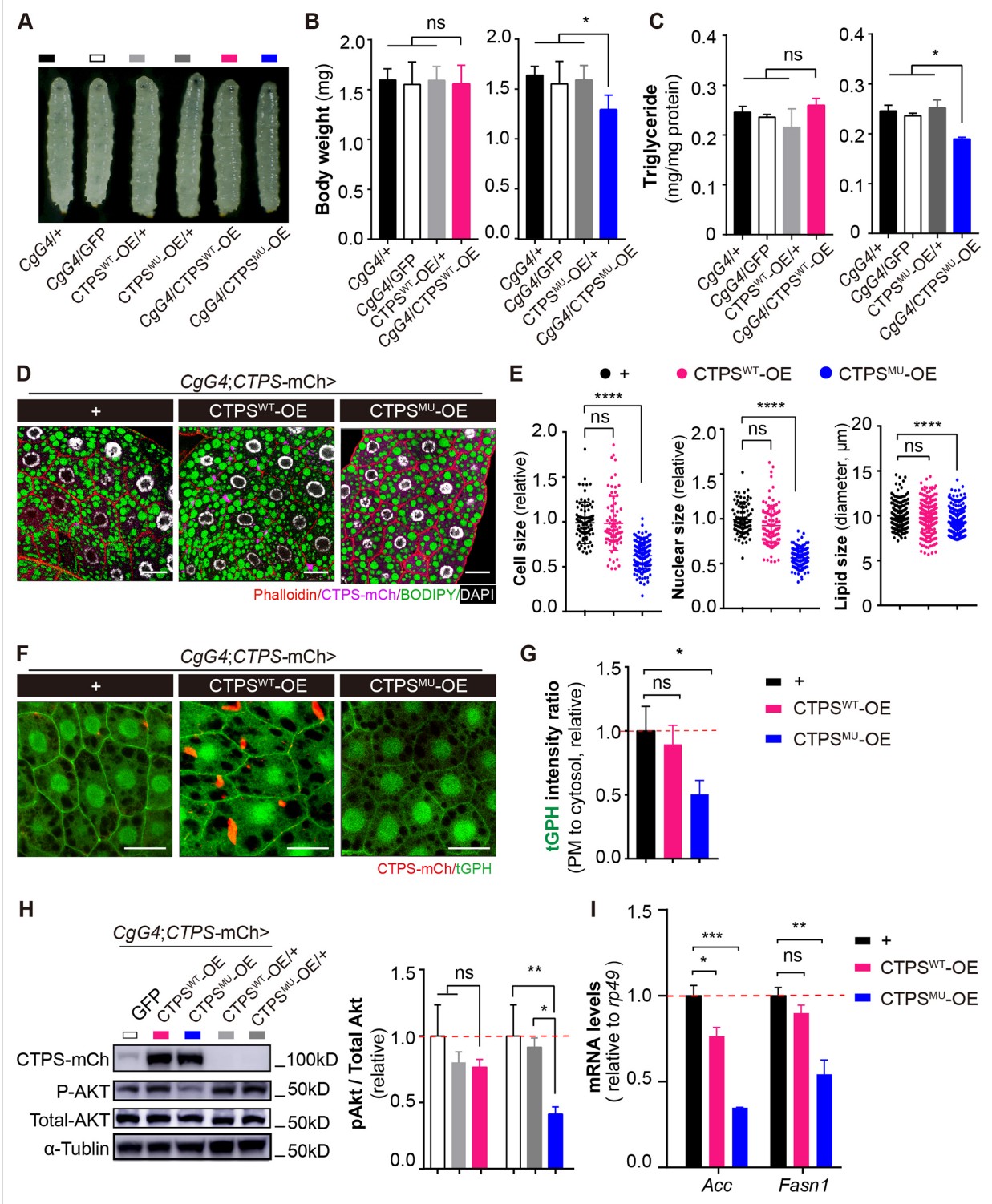

**Figure 6.** Disrupting the filament-forming property of CTPS alleviates HFD-induced obesity. (**A**) Representative photograph of HFD-fed 76~80 hour AEL larvae showing larval morphology. The *CgG4>* CTPS^MU OE line is compared with *CgG4>*+, *CgG4>*GFP, *CgG4>*CTPS^WT-OE, CTPS^WT-OE/+, and CTPS^MU-OE/+ lines. (**B**) Body weights of 76~80 hour AEL larvae (30 larvae/group; 5–6 groups/genotype; 3 biological replicates). The *CgG4>*CTPS^WT-OE line is compared with CTPS^WT-OE/+, CTPS^WT-OE/+, *CgG4>*GFP, and *CgG4>*+ lines. The *CgG4>*CTPS^MU-OE line is compared with *CgG4>*+, *CgG4>*GFP, and CTPS^MU-OE/+ lines. (**C**) TAG concentrations in 76~80 hour AEL larvae. TAG concentrations are normalized to total protein concentration (10 larvae/group; 5–6 groups/genotype; 2 biological replicates). (**D**) Representative confocal images of fat bodies from HFD-fed 76~80 hour AEL larvae. Fat bodies are stained with phalloidin (red) to reveal the cell outline, BODIPY493/503 (green) to reveal lipid droplets, and DAPI (white) to reveal nuclei.

*Figure 6 continued on next page*

Figure 6 continued

The fly lines are *CgG4; CTPS*-mCh >+*CgG4;CTPS*-mCh >CTPS^WT-OE, and *CgG4;CTPS*-mCh >CTPS^MU -OE. Scale bars, 30 µm. (**E**) Quantification of cell size, nuclear size, and lipid droplet size from (**D**). Values are normalized to the control line *CgG4;CTPS*-mCh>+ (10 images/genotype; 3 biological replicates). (**F**) Representative confocal images of PI3K activation in the fat bodies of 3^rd instar larvae. The membrane location of tGPH (green) shows the activity of PI3K. Scale bars, 10 µm. (**G**) tGPH intensity ratio of the cell membrane relative to the cytosol from the images in (**F**). The values are normalized to the control (10–15 images/genotype; 3 biological replicates). (**H**) Western blot analysis of phosphorylated Akt from fat body lysates. Anti–mCh, anti-phosphorylated-Akt, and anti-total-Akt antibodies were used for the immunoblotting analysis. Alpha-tubulin was used as an internal control. The P-Akt to total-Akt ratio is shown in the right panel. The values are normalized to the control line *CgG4;CTPS*-mCh>GFP (3 biological replicates). (**I**) Quantitative RT-PCR analysis of *acc* and *fasn1* mRNA abundances in the fat body lysates of 76~80 hour AEL larvae (30 larvae/group; 4 groups/ genotype; 2 biological replicates). All data are shown as mean ± S.E.M. ns, not significant, * $P<0.05$, ** $P<0.01$, *** $P<0.001$, and **** $P<0.0001$ by Student's *t*-test or one-way ANOVA with a Tukey *post hoc* test.

The online version of this article includes the following source data and figure supplement(s) for figure 6:

**Source data 1.** Uncropped gel of phosphorylated Akt from fat body lysates.

**Figure supplement 1.** Quantitative RT-PCR analysis.

**Figure supplement 2.** Quantitative RT-PCR analysis.

## Discussion

Obesity has become an epidemic disease globally, affecting over 1.9 billion adults who were overweight and 650 million who were considered obese as of 2016. In addition, in 2020, approximately 39 million children under 5 were either overweight or obese. *Drosophila* has emerged as a preferred model for investigating lipid metabolism and homeostatic regulation (*Birse et al., 2010*; *Baker and Thummel, 2007*) because of its evolutionarily and functionally conserved metabolic signaling pathway (*Zhang et al., 2020*). Our study uncovered a relationship between CTPS cytoophidia and lipid homeostasis in the context of HFD-induced obesity. Our results demonstrate that the lack of CTPS impairs the growth of adipocytes and the accumulation of lipids in response to HFD consumption, probably by inhibiting the PI3K-Akt-SREBP signaling pathway.

CTPS is a metabolic enzyme that catalyzes the rate-limiting step in the de novo synthesis of CTP, which is vital for the synthesis of RNA, DNA, and phospholipids in cells. Its potential functions in cells and developmental biology are being studied due to its importance in nucleotide and phospholipid production. In yeast cells, a mutation in the *URA7* gene, which encodes CTPS, leads to increased production of CTP, resulting in a higher synthesis rate for phosphatidate and PC (*Ostrander et al., 1998*; *Chang and Carman, 2008*). CTPS1 and its interaction with ENDU-2 regulate germ cell proliferation and nucleotide metabolism in *Caenorhabditis elegans* (*Liss et al., 1982*). Recent studies have shown that CTPS can form filamentous structures in prokaryotic and eukaryotic cells (*Liu, 2010*; *Ingerson-Mahar et al., 2010*; *Noree et al., 2010*), and that when polymeric cytoophidia form, CTPS enzymatic activities in *E. coli* are suppressed (*Barry et al., 2014*). Some findings have led to the assumption that filament formation boosts catalytic activity in human CTPS (*Lynch et al., 2017*; *Strochlic et al., 2014*), contrary to the previous assumption that filaments contain only inactive enzymes. Despite the fact that species differ in their ability to form filaments, recent studies suggest that CTPS filaments can dynamically switch between active and inactive states in response to changes in substrate and product levels, resulting in different regulatory consequences (*Zhou et al., 2019*; *Lynch and Kollman, 2020*). Filament formation has been shown to increase protein stability (*Sun and Liu, 2019*), and cytoophidia in *Drosophila* ovaries have been found to have elevated levels of enzymatic activity (*Strochlic et al., 2014*). Our research has shown that CTPS cytoophidia are crucial for integrin-Collagen IV-mediated adipocyte adhesion (*Liu et al., 2022*) and for the proliferation of *Drosophila* intestinal stem cells (*Zhou et al., 2021b*; *Zhou et al., 2022*). Several oncogenes, including Myc, Ras, the ubiquitin E3 ligase (Cbl), as well as activated CDC42-associated kinase (Ack), modulate the formation of CTPS cytoophidia, suggesting that cytoophidia play a role in regulating cell growth and metabolic balance in various contexts (*Strochlic et al., 2014*; *Zhou et al., 2022*; *Aughey et al., 2016*; *Wang et al., 2015*).

The fat body is an essential organ that regulates lipid accumulation in response to HFD feeding in *Drosophila*. The growth and remodeling of white adipose tissue (WAT) have a direct impact on developing metabolic syndrome in obesity, with healthy WAT growth characterized by smaller and more adipocytes (*Vishvanath and Gupta, 2019*; *Ghaben and Scherer, 2019*). CTPS is abundantly produced

in mammalian adipose or hepatic tissues. Our study revealed that HFD feeding increased CTPS transcription and cytoophidia elongation in the *Drosophila* fat body. This suggests that cytoophidium formation is a dynamic process, and that CTPS cytoophidia can respond to changes in nutrient availability. This is supported by the observation of an elevated abundance of CTPS cytoophidia in human cancer tissues, including hepatocellular carcinoma (*Chang et al., 2017*). Our findings further showed that CTPS plays a critical role in regulating systemic energy homeostasis, mainly in the fat body, as CTPS deficiency in adipose tissue led to decreased body weight and increased susceptibility to starvation during food deprivation. Moreover, the absence of cytoophidia, as the result of reduced CTPS expression or the expression of a dominant-negative mutant CTPS protein that prevents the polymerization, led to a reduction in HFD-induced adipocyte expansion and TAG level, strongly supporting the role of cytoophidia in adipocyte growth and lipogenesis in response to HFD.

This study showed that the appropriate regulation of lipid metabolism in *Drosophila* adipocytes requires the presence of CTPS. Inhibition of CTPS, through either RNAi or the expression of a mutant form (H355A), results in the suppression of PI3K activity and a decrease in the level of phosphorylated Akt. Akt activates SREBP, which stimulates lipid synthesis, including the production of fatty acids (*Krycer et al., 2010*; *Yecies et al., 2011*; *Yi et al., 2020*). In *Drosophila*, there is only one homolog of SREBP, which regulates lipid metabolism genes such as *Acc* and *Fasn1* (*Seegmiller et al., 2002*). *Fasn1* is the key metabolic multi-enzyme critical to the terminal step of fatty acid synthesis. In humans, genetic variation within *FASN* is associated with obesity (*Kovacs et al., 2004*), and high transcriptional activation of *FASN* occurs in cancer cells (*Menendez et al., 2005*; *Menendez and Lupu, 2007*; *Baron et al., 2004*). In both *Drosophila melanogaster* and human cells, activation of SREBP contributes to Akt-dependent cell growth (*Yecies et al., 2011*; *Porstmann et al., 2008*). Our data indicate that CTPS is involved in the cell-autonomous regulation of adipocyte growth by maintaining the activation of the PI3K-AKT-SREBP signaling pathway. Moreover, fat body growth relies on endoreplication, a process through which DNA replication occurs without cell division, leading to an increase in cell size and polyploidy. This process is crucial for the accumulation of biomass, which increases cellular volume and organelle content, both of which are crucial for TAG storage. Thus, the loss of CTPS, which impairs nucleotide synthesis, could affect endoreplication and could ultimately reduce fat storage capacity. Our data clearly indicate that CTPS plays a crucial role in coordinating these cellular processes in the fat body.

In summary, our study provides evidence for the essential role of CTPS in regulating adipocyte growth and lipid metabolism in *Drosophila* through the activation of the PI3K-Akt-SREBP pathway. Further investigations are necessary to determine whether this mechanism is also present in mammalian adipose tissue and in other tissues, such as liver. A more comprehensive understanding of the interplay between diverse cellular processes in maintaining lipid homeostasis could lead to the advancement of knowledge regarding metabolic disorders and energy homeostasis.

## Materials and methods
### Generation of transgenic flies
CRISPR/Cas9 technology was used to establish the C-terminal mChe-4V5 tagged CTPS knock-in fly according to homology-directed repair procedures previously described by researchers at Fungene Biotech (*Liu et al., 2022*; *Bassett et al., 2013*; http://www.fgbiotech.com).

To generate transgenic UAS-CTPS and UAS-CTPS^MU flies, the cDNAs encoding *Drosophila CTPS* were produced by RT-PCR using the total RNAs extracted from the *w^1118* line (#3605, from the Bloomington *Drosophila* Stock Center), as previously described by researchers at the Core Facility of *Drosophila* Resource and Technology, SIBCB, CAS (*Liu et al., 2022*; *Ni et al., 2008*). The transgenic lines were backcrossed into the *w^1118* background for over five generations before further genetic manipulations.

### Fly strains
The GAL4/UAS system (*Brand and Perrimon, 1993*) was utilized for adipocyte-specific expression or RNAi knockdown of the desired genes. The fly lines were obtained from the Bloomington *Drosophila* Stock Center (Department of Biology, Indiana University, Bloomington, IN); they included *w^1118* (stock number 3605), *Cg GAL4* (stock number 7011), *ppl GAL4* (stock number 5092), *CTPS*-RNAi[TRiP.JF02214]

(stock number 31924), *CTPS*-RNAi[TRiP.HM04062] (stock number 31752), UAS-SREBP.Cdel (stock number 8243), and tGPH (stock number 8163). The UAS-GFP fly line (stock number THJ0079) was from the TsingHua Fly Center (TsingHua University). The RNAi control (Con-Ri, stock number, V60101) line was from the Vienna *Drosophila* Stock Center (*Dietzl et al., 2007*). The *Tubulin GAL4, tubulin GAL80*[ts] was kindly gifted by Dr Margaret Su-chun Ho of ShanghaiTech University. The *yw,hsflp;act>CD2>GAL4*, UAS-GFP line was kindly gifted by Dr Lei Zhang of Shanghai Jiao Tong University.

## Fly husbandry and diet preparation

Fly lines were cultured on standard yeast-cornmeal-agar food comprising 250 g yeast, 92 g soy flour, 668 g cornmeal, 400 g sucrose, 420 g maltose, 60 g agar, 25 g methylparaben (dissolved in 95% ethanol), 10 g sodium benzoate, and 68 ml propionic acid in 10 liters of de-ionized water. To prepare a high-fat diet, we added coconut oil (30% v/v) and yeast (29.5 g/L) to the standard diet and mixed it thoroughly.

To ensure that the larvae used in our study were at the appropriate developmental stage, we collect embryos for a maximum of 4 hours. To achieve similar larval densities between control and experimental lines, we transferred 80 embryos to vials containing regular or high-fat food. For body weight measurements or starvation assays, newly eclosed flies were collected and kept in standard fly food bottles (~200 flies per bottle) for 5 days. During this time, mating was allowed. All experiments were conducted under a 12 h/12 h light/dark cycle at 25 °C with 50% humidity.

## Fly body weight

The embryos were collected for 2–4 hours and subsequently fed with the experimental diets. When the embryos reached 76–80 hours after egg laying (AEL), the larvae were washed in PBS and weighed to determine their body weight (30 flies per group; 3–6 groups per genotype). Five-day-old adults were measured for body weight (30 flies per group; 3–6 groups per genotype). Body weight gain was calculated using the following formula:

$$\Delta BW : \text{Body weight gain}$$

$$\Delta BW = BW_{HFD} - BW_{RD}$$

$$\text{Reduction of body weight gain} = \frac{(\Delta BW_{con} - \Delta BW_{CTPS-Ri})}{\Delta BW_{con}} \times 100\%$$

## Larvae size

3[rd] instar wandering larvae were rinsed with a PBS solution and subsequently microwaved in the same solution for 5–10 seconds to make them rigid before being photographed. Their length was measured using FIJI-ImageJ software, and the experiments were conducted at least three times.

## Starvation assay

Male and female fruit flies (separately), aged five days, were sorted into vials containing 3 ml 1% agar: ~30 flies per vial; 5–6 groups per genotype. The flies were transferred to new vials every two days to prevent bacterial contamination. The number of dead flies was recorded every 12 hours until all of the flies had died.

## Floating assay

The floating assays were performed with slight modification of a previously described protocol (*Reis et al., 2010*). Briefly, we placed ten 3[rd] instar wandering larvae into a vial containing 10 ml of a 12% sucrose phosphate-buffered saline (PBS) solution. The number of larvae floating at the surface of the solution was counted within 3 minutes. The data are presented as the percentage of floating larvae, and the experiments were conducted at least three times.

## Triglyceride analysis

Samples for the TAG concentration assay were obtained by snap-freezing 76~80 hour AEL larvae or male adults in liquid nitrogen and storing them at –80 °C. Each biological replicate comprised 6–10 flies collected into a 1.5 ml microcentrifuge tube, and each experiment included 3–6 biological replicates for each genotype. The triglyceride concentration was determined using a coupled colorimetric

assay, as previously described (*Liu et al., 2012*). Briefly, samples were homogenized in PBS containing 1% Triton X-100 and incubated at 70 °C for 10 min. After that, the homogenates were incubated with Triglyceride Reagent (Sigma, T2449) for 60 min at 37 °C, followed by incubation with Free Glycerol Reagent (Sigma, F6428) for 5 min at 37 °C. The samples were assayed using a microplate spectrophotometer at 540 nm, and TAG levels were normalized to the protein level.

## Immunohistochemistry

To perform immunofluorescence staining, we dissected the fat bodies from the larvae in Grace's Medium and fixed them in 4% formaldehyde in PBS for 15 min. For membrane staining, we washed fixed fat bodies twice for 5 min in PBS, and then incubated them with 0.165 µM Alexa Fluor 488, 568, or 633 phalloidin (Invitrogen) in PBSTG (PBS +0.5% Triton X-100 +5% normal goat serum) for 30 min at room temperature (RT). Then, samples were rinsed in PBS twice for 5 min each time and mounted on a Vecta shield with DAPI (Invitrogen). For lipid droplet staining, we washed fixed fat bodies twice for 5 min in PBS, and then incubated them with Nile red (10 µg/mL for 30 min at RT) or BODIPY 493/503 (1 µg/mL for 30 min at RT). The samples were then rinsed in PBS twice for 5 min each time and mounted on a Vecta shield with DAPI (Vector Labs).

## Mosaic analysis

We used the *hs-Flp; Act>CD2>Gal4*/UAS system to generate clones in larval fat body cells. 24 hr after egg deposition, we induced the transgenes for 30 min at 37 °C. We then dissected the fat bodies from 3rd instar larvae and fixed them in 4% formaldehyde in PBS for 15 minutes. To analysis the sizes of the cell in the clones, larval fat body cells were stained with 0.165 µM Alexa Fluor 488 phalloidin (Invitrogen) in PBSTG (PBS +0.5% Triton X-100 +5% normal goat serum) for 30 min at RT. After that, samples were rinsed in PBS twice for 5 min each time and mounted on a Vecta shield with DAPI (Invitrogen).

## Imaging and image analysis

Fluorescent images were obtained using confocal laser scanning microscopy (Leica SP8) at 20 X, 40 X, or 63 X for oil objects. To compare the sizes of cells, nuclei, and lipid droplets, the images of the central focal/z section with the largest nucleus were collected. Approximately 250 adipocytes from each genotype were measured to quantify cell and nuclear size using FIJI-ImageJ. For lipid droplet size, we measured the diameter of lipid droplets (those larger than 4 µm that can be accurately measured) in approximately 250 fat cells from RD-fed larvae or in approximately 100 fat cells from HFD-fed larvae, using FIJI-ImageJ. We counted the length and number of CTPS cytoophidia in cells by analyzing 40 X confocal images using FIJI-ImageJ. The data were normalized by the number of cells in one image. To quantify tGPH signal, we utilized Cellpose (*Stringer et al., 2021*), a deep learning-based segmentation method, to segment cell membrane contours with a diameter of 150 pixels using the Cytoplasm model. The resulting binary images of the outlines were then dilated to 8 pixels to create membrane masks in FIJI-ImageJ. In Imaris 8.0, two channels were analyzed: channel A for tGPH and channel B for the cell membrane mask. The fluorescent intensity of tGPH was obtained using Imaris' intensity-based coloc methods. The data represent the ratio of the cell membrane fluorescent intensity to the cytosolic fluorescent intensity in cells.

## Western blot

Larval fat body tissues were homogenized in RIPA buffer (150 mM NaCl, 1% NP-40, 0.5% sodium deoxycholate, 0.1% SDS, 50 mM Tris–HCl, pH 7.4) using a Tissuelyser-24 grinder (Jingxin, Shanghai, China). After centrifugation at 15,000 g at 4 °C for 20 min, the supernatants were subjected to separation by SDS-PAGE before immunoblotting analysis. The primary antibodies used are rabbit anti-phospho-Akt (Ser473) antibody (1:1000, Cell Signaling, Catalogue no.9271s), rabbit anti-total-Akt antibody (1:1000, Cell Signaling, Catalogue no.9272), mouse anti-mCherry (1:3000, Abbkine, Catalogue no. A02080), mouse anti-α-tubulin antibody (1:4000; Sigma, Catalogue no. T6199). Anti-secondary antibodies are anti-rabbit IgG (1:2000, Cell Signaling, Catalogue no. 5151) and horseradish peroxidase (HRP)-conjugated anti-mouse IgG (1:2000, Cell Signaling, Catalogue no. 7076). Non-saturated bands were quantified on FIJI-ImageJ (National Institutes of Health) and presented as a ratio in relation to total-Akt. At least three biological replicates were quantified.

## Lipidomic analysis

Lipids were extracted from early 3rd instar larval fat bodies as previously described (*Lam et al., 2022*). The lipidomic analyses were carried out on an ExionLC-AD system coupled with a Sciex QTRAP 6500 PLUS system. The separation of individual classes of polar lipids by normal phase HPLC was carried out using a TUP-HB silica column (i.d. 150x2.1 mm, 3 µm) with the following conditions: mobile phase A (chloroform:methanol:ammonium hydroxide, 89.5:10:0.5) and mobile phase B (chloroform:methanol:ammonium hydroxide:$H_2O$, 55:39:0.5:5.5). MRM transitions were set up for quantification by referencing spiked internal standards. The mixed internal standard includes $d_9$-PC32:0 (16:0/16:0); $d_7$-PE33:1 (15:0/18:1); $d_{31}$-PS (d31-16:0/18:1); $d_7$-PA33:1 (15:0/18:1); $d_7$-PG33:1 (15:0/18:1); $d_7$-PI33:1 (15:0/18:1); $d_5$-CL72:8 (18:2)4; $d_7$-LPC18:1; $d_7$-LPE18:1; $C_{17}$-LPI; $C_{17}$-LPA; $C_{17}$-LPS; and $C_{17}$-LPG (Avanti Polar Lipids). Free fatty acids were quantitated using $d_{31}$-16:0 (Sigma-Aldrich).

## RNA isolation and RNA sequencing

Briefly, total RNA was isolated from the dissected fat bodies of 50 to 100 2nd instar larvae using TRIzol reagent (Invitrogen) according to the manufacturer's recommendations. The libraries were constructed using the TruSeq Stranded mRNA LT Sample Prep Kit (Illumina, San Diego, CA, USA) according to the manufacturer's instructions. The thresholds for significantly different expression were set at $P<0.05$ and a fold change greater than 2 or less than 0.5. Transcriptome sequencing and analysis were conducted by OE Biotech Co., Ltd. (Shanghai, China). The raw experimental data have been deposited in the Gene Expression Omnibus database at the National Center for Biotechnology Information, and can be accessed using the identifier GSE221707.

## Quantitative RT-PCR

Total RNAs were prepared from larval fat bodies, larval whole body, or adult flies using the TRIzol reagent (TransGen Biotech, Beijing, China). cDNAs were synthesized with PrimeScript RT Master Mix (Takara), followed by the addition of template RNA. 2 X SYBR Green PCR Master Mix was purchased from Bimake. Real-time quantitative PCR was conducted using the QuantStudion 7 Flex System (Applied Biosystems). For normalization, *actin*, *rp49*, or *rp32* was utilized as the internal control. The oligonucleotide primers used were as follows:

> *rp49*: sense 5'-TCCTACCAGCTTCAAGATGACC-3', antisense 5'-CACGTTGTGCACCAGGAACT-3';
>
> *CTPS*: sense 5'-GAGTGATTGCCTCCTCGTTC-3', antisense 5'-TCCAAAAACCGTTCATAGTT-3'.
>
> *Acc*: sense 5'-GTGCAACTGTTGGCAGATCAGTA-3', antisense 5'-TTTCTGATGACGACGCTGGAT-3'
>
> *Fasn1*: sense 5'-CCCCAGGAGGTGAACTCTATCA-3', antisense 5'- TTTCTGATGACGACGCTGGAT-3'
>
> *Srebp:* sense 5'-GGCAGTTTGTCGCCTGATG-3', antisense 5'-CAGACTCCTGTCCAAGAGCTGTT-3'
>
> *Scap:* sense 5'-ACCAGAGCAGCGAAAACAAAC-3', antisense 5'- GAGAGTTCTGCGTCCACAGG-3'
>
> *nmdyn-D6*: sense 5'-GAGCCCTGATCTCCCAGAAC-3', antisense 5'- TAGCTGGGTCCGCTGTTCAT-3'
>
> *nmdyn-D7*: sense 5'-GACGGATGTCTCCTCTTCAGTC-3', antisense 5'- TCTTCCAAACTGGGCGACAG-3'
>
> *CG15547*: sense 5'-GGGGTTTATGCTGGAGGTCA-3', antisense 5'- TCCGATGCCGAACCAAAATAA-3'
>
> *Pi3K21B*: sense 5'-AGGAGCACAAGCAGACACTC-3', antisense 5'-ATCCTTTTAGGCGCTCAATGT-3'
>
> *Pi3K59F*: sense 5'-GCAAATCAAGGTAGGGACGC-3', antisense 5'-GCCTTGTAGGAGCTGGTCAC-3'
>
> *Pi3K68D*: sense 5'-TGCTAAACGACAATACTGGCAAC-3', antisense 5'-CCACCTGTTGACTGCCTCA-3'
>
> *Pi3K92E*: sense 5'-TGGATAGCAAGATGCGACCG-3', antisense 5'-TGCGGAAGTCCATACCATCG-3'
>
> *rp32*: sense 5'-GCTAAGCTGTCGCACAAATG-3', antisense 5'- GTTCGATCCGTAACCGATGT-3'

*actin5c*: sense 5'-ATTTGCCGGAGACGATGCTC-3', antisense 5'-CCGTGCTCAATGGGGTACTT -3'

*eGFP*: sense 5'-CCCGACAACCACTACCTGAG-3', antisense 5'- GTCCATGCCGAGAGTGATCC -3'

## Statistical analysis

All data are presented as the mean ± standard error of the mean (S.E.M.) for at least 2–3 independent experiments. Statistical comparisons of each genotype and the controls were performed by a unpaired two-tailed Student's *t*-test, whereas comparisons between multiple genotypes were performed by one-way or two-way ANOVA with a Tukey *post hoc* test in GraphPad Prism 7.0. *P*<0.05 was considered to be statistically significant.

## Acknowledgements

We thank Xiaoming Li from the Molecular Imaging Core Facility (MICF) at the School of Life Science and Technology at ShanghaiTech University for providing technical support. We also thank the Core Imaging Facility at the National Center for Protein Science Shanghai (NCPSS), the Core Facility of *Drosophila* Resource and Technology at SIBCB, CAS, and the online database flybase (http://flybase. org) (*Gramates et al., 2022*). We acknowledge the Bloomington *Drosophila* Stock Center (NIH P40OD018537) and the Vienna *Drosophila* Resource Center (VDRC, https://www.vdrc.at) at Vienna BioCenter Core Facilities, part of the Vienna BioCenter, Austria for providing *Drosophila* stocks used in this study. This work was supported by grants from the Ministry of Science and Technology of China (no. 2021YFA0804701-4), the National Natural Science Foundation of China (no. 32071144), and the Shanghai Science and Technology Commission (no. 20JC1410500) to JL and J-LL.

## Additional information

### Funding

| Funder | Grant reference number | Author |
| --- | --- | --- |
| Ministry of Science and Technology of the People's Republic of China | 2021YFA0804701-4 | Ji-Long Liu |
| National Natural Science Foundation of China (32071144) | | Jingnan Liu |
| Shanghai Science and Technology Commission | 20JC1410500 | Ji-Long Liu |

The funders had no role in study design, data collection and interpretation, or the decision to submit the work for publication.

### Author contributions

Jingnan Liu, Conceptualization, Formal analysis, Funding acquisition, Investigation, Writing – original draft, Writing – review and editing, Supervision, Methodology, Project administration, Resources; Yuanbing Zhang, Qiao-Qi Wang, Youfang Zhou, Investigation; Ji-Long Liu, Conceptualization, Supervision, Funding acquisition, Writing – review and editing

### Author ORCIDs

Jingnan Liu ⓘ http://orcid.org/0000-0002-4325-1796
Yuanbing Zhang ⓘ http://orcid.org/0000-0002-4569-2798
Qiao-Qi Wang ⓘ http://orcid.org/0000-0001-6580-2969
Youfang Zhou ⓘ http://orcid.org/0000-0001-6088-765X
Ji-Long Liu ⓘ http://orcid.org/0000-0002-4834-8554

### Decision letter and Author response

Decision letter https://doi.org/10.7554/eLife.85293.sa1

Author response https://doi.org/10.7554/eLife.85293.sa2

## Additional files

### Supplementary files
• MDAR checklist

### Data availability
Sequencing data have been deposited in GEO under accession codes GSE221707. All data generated or analysed during this study are included in the manuscript and supporting file; Source Data files have been provided for Figures 5, Figure 6, and Figure 5-figure supplement 1.

The following dataset was generated:

| Author(s) | Year | Dataset title | Dataset URL | Database and Identifier |
|---|---|---|---|---|
| Liu J, Zhang Y, Liu J | 2023 | Fat body-specific reduction of CTPS alleviates HFD-induced obesity | https://www.ncbi.nlm.nih.gov/geo/query/acc.cgi?acc=GSE221707 | NCBI Gene Expression Omnibus, GSE221707 |

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
