## [Editor Report]

This study describes a role for CTPS (Cytidine 5'-triphosphate synthase) and CTPS filamentous structures (cytoophidia) in regulating fat storage in the fly fat body in normal and high-fat diets. The data were collected and analyzed using validated, solid methodologies. These results are useful for biologists interested in general cellular mechanisms of metabolism.

---

## [Decision Letter]

**Decision letter after peer review:**

[Editors’ note: the authors submitted for reconsideration following the decision after peer review. What follows is the decision letter after the first round of review.]

Thank you for submitting the paper "Fat body-specific reduction of CTPS alleviates HFD-induced obesity" for consideration by *eLife*. Your article has been reviewed by 3 peer reviewers, one of whom is a member of our Board of Reviewing Editors, and the evaluation has been overseen by a Senior Editor. The reviewers have opted to remain anonymous.

Comments to the Authors:

Your manuscript has been extensively reviewed by 3 reviewers with a depth of relevant expertise in your area. The reviewers found your work to be potentially exciting but in the process of review they also identified a number of issues. Unless these issues are addressed we cannot accept your paper for publication in *eLife*. Please note this will require additional experimental investigation and it is critical that you address these issues if you wish the work to be further considered at *eLife*. In view of the scope of the comments raised we are rejecting the paper for now but if you decide to address the multiple points brought up by the reviewers, we would be happy to reconsider this decision in a future submission.

Specifically, the potential confounding effects of changes in size and growth, and how these affect lipid levels should be addressed. In general more rigorous experimental conditions need to be implemented keeping in consideration controls including genetic backgrounds. Generally, for lifespan assay, lines must be in the same background and must have been isogenized for multiple generations. The different assays also need to take into account how reporters, growth and developmental conditions are controlled for as these can affect results.

*Reviewer #1 (Recommendations for the authors):*

In this manuscript Liu and colleagues describe a role for the CTPS enzyme and cytoophidia with potentially functions in the fly fat body. The authors propose that CTPS and cytoophidia have a detrimental role in animals fed a high fat diet. The authors go on to propose that depletion of CTPS in HFD conditions reduces the number of cytoophidia formed and this results in a protective role against HFD-induced obesity. I find the work highly interesting and exciting, but somewhat preliminary. The target audiences would be those in the areas of metabolism and general cellular biology.

At this stage, several issues require additional consideration, including: how changes in growth might contribute to the observed phenotypes; the specific effects of high-fat diet versus other diets; changes in lipids and phospholipids versus the direct effects of CTPS and cytoophidia. There are instances where controls for experiments are missing, compromising the ability to interpret the data as fully supporting the conclusions. Additionally, alternative approaches and/or conclusions have not been considered. Similarly, potential implications for such phenomena are not presented or discussed in the context of what is already known.

All data needs to be revised with more and better controls. Genetics and growth conditions play a strong influence in metabolic outcomes. Therefore, for assurance that some of the results are not just due to these but indeed to the experimental manipulations, the following major points need to be addressed:

1.To rule out genetic contributions from insertions, each and all experiments need the genetic background control for each line, not just the drivers but also the RNA-i lines and UAS-overexpression lines. Each experiment also needs an overexpression control (GFP or similar) or a RNAi control (RFP-RNAi or similar). This is specifically requested for the all the life-span experiments, density assays and TAG measurements

2. Methods state that flies are allowed to lay for 4 hours, but don't state if density for each genotype in each vial was controlled for. Overcrowding or underpopulation of animals per vial/bottle can influence their ultimate size, food intake and metabolism. Were number of animals per vial controlled for? If not, please address.

3. What are the levels of knockdown for each one of the RNAi lines? What are the levels of expression from each one of CTPS wt and mutant lines? Levels of knockdown and overexpression per driver could account for the differences in phenotype observed. Could you please show this is not the case?

4. For evaluation of AKT activity via phosphorylation, an independent and quantifiable western blot for the fat bodies shown in Figure 5F,G and Figure 6G,H is requested.

5. tGPH was developed to be used to measure insulin signaling by taking into account the amount of PIP3 in the nucleus versus the membrane. Total fluorescence is not commonly used as a reporter for INR signaling activity. Please either use the reporter as it was developed to be used, or demonstrate that total fluorescence can be used as an accurate reporter.

6. for figures 3A, 4A,C 5D,F, 6 D,E and G, some of these images seem to represent a more apical view of fat bodies than others. In order to compare levels of lipid droplets, tGPH, etc, images should be of a comparable central focal plane/z section, otherwise, lipid droplets can seem bigger or smaller, and nuclei fainter or brighter. Could you provide similar sections for each image?

7. In figure 4, it seems that larval size changes with the CTPS-RNAi conditions. This would lead to different outcomes in terms of lipid levels and fat body sizes. Could you please measure larval size so that size as a variable is taken into account?

8. Please use additional, and more than one, endogenous controls for the qPCR, as there seems to be a size difference in the RNAi larvae (above). RP49 is a ribosomal protein that correlates with cell size, and thus is not appropriate as the qPCR endogenous control.

9. In figure 4, why are levels of GFP lower in the RNAi condition? This is not expected, as the GFP should still be labelling the fat cells that are present.

10. In the Discussion, please discuss what is known about CTPS and cytoophidia in the context of lipid metabolism and growth regulation. What role do the authors imagine in the fly fat body and in response to diet? Is this role expected to be high-fat-diet-specific? What would be the role of the cytoophidia? And how would this role lead to regulation of the expression of lipogenic genes?

11. CTPS regulation of lipid metabolism and phospholipids: do the observed effects result from accumulation of CTPS products, or due to the formation of cytoophidia per se?

*Reviewer #2 (Recommendations for the authors):*

In this manuscript Liu et al. determine that CTP synthase (CTPS) is required in the fat body to regulate TAG levels and body weight. The authors also show that CTPS transcript levels are more abundant on a high fat diet and propose that manipulation of CTPS specifically in the fat body rescues defects observed. The authors perform RNA sequencing of fat bodies from control and CTPS RNAi larvae and determine that genes involved in lipogenesis are significantly decreased. The authors also provide fluorescence analysis that CTPS may interact with the PI3K pathway to regulate lipid levels. The authors suggest that CTPS cytoophidia regulate induction of high fat diet related obesity.

The initial observation that CTPS regulates lipid levels in the fat body through regulation of PI3K signaling. However, it is unclear how the authors came to some of their conclusions and how their results significantly enhance findings from prior studies characterizing this protein and the cytoophidia that form.

1. It is unclear whether transgene induction occurred during development for their analysis of adults (Figure 1). Also, did the authors maintain the flies for RNAi at 25 degrees or shift them to 29 degrees?

2. Could the authors provide reasoning as to why global knockdown of CTPS using tubulin-Gal4 gives a relatively weak phenotype in body weight compared to fat body-specific RNAi (Figure 1A and C)?

3. Why do the authors propose that there is a decrease in starvation resistance when CTPS is knocked down in the central nervous system using Elav-Gal4, but not a change in body weight (Figure 1B and E)?

4. The authors should address the fact that the Cg-Gal4 driver used in their study is shut off by knockdown of CTPS in both a regular and high fat diet. For example, using Cg-Gal4 in combination with UAS-GFP is an indication of driver activity. The authors should provide reassurance to the readers that they are getting sufficient knockdown of CTPS by qRT-PCR and provide some explanation as to how they can make comparisons/conclusions between regular and high fat diets. This manipulation can alter the conclusions made in their studies going further using this set up. It may be beneficial to do a time course to determine how long after transgene induction that this switch occurs. Alternatively, the authors could determine if PPL-Gal4 would be a more suitable driver for these experiments. Furthermore, the authors should ensure that the driver remains active in their overexpression analysis with wild-type and the H355A mutant.

5. The authors propose that the GFP level fluorescence increase using the Cg-Gal4 driver on a high fat diet suggests increased lipogenesis (line 171). However, this observation is not quantified and there is not any evidence that the Cg-Gal4 driver can be used as a readout of lipid levels. This driver just represents a regulatory region between the Collagen IV genes (vkg and Cg25C – see Asha et al. 2003).

6. How is body weight gain determined (line 179-180)?

7. Some of the images provided in the manuscript do not look like the fluorescence intensity has been scaled between the control and experimental images equivalently. For example, in Figure 4C the regular diet control and CTPS RNAi lipid droplet intensity does not look the same. Even if there are fewer and smaller, the BODIPY staining intensity in general should be uniform.

8. The authors propose that CTPS regulates PI3K signaling and show activity using reporters and antibody. Given the number of tools in the *Drosophila* community, this study would be greatly enhanced if genetic interaction experiments were performed.

9. The authors should change loss of CTPS inhibits SREBP transcriptional activity (lines 221-222) to inhibits SREBP transcript levels.

10. It is unclear why there is an increase in body weight with overexpression of wild-type CTPS, but not an increase in TAG levels (Figure 6B). Could the authors please provide an explanation?

11. The authors claim that lipid droplet accumulation and size are changed with overexpression of WT and the H355A CTPS mutant, but do not provide any quantification (Figure 6D).

12. The authors should provide the full recipe for their regular food diet as there is not a "standard yeast-cornmeal-agar food" (line 380).

13. What is PBSTG (line 427)? Please provide the recipe.

14. The manuscript should be further edited for clarity. For example, there are quite a few run-on sentences that are difficult to follow. In addition, summary statements/conclusions of the data in the Results section (as done for lines 189-190) would be helpful to the reader.

*Reviewer #3 (Recommendations for the authors):*

This work investigates the role of CTP synthase (CTPS) in the control of triglyceride storage in response to normal and high fat diet feeding in *Drosophila melanogaster*. CTPS generates the nucleotide CTP, a molecule required for phospholipid, RNA and DNA synthesis. CTPS is regulated, in part, by its ability to form filaments called cytoophidia that promote enzyme activity. Using loss-of-function approaches in *Drosophila*, the authors find that CTPS is required for growth and lipid accumulation in flies in normal conditions and in response to high fat diet feeding, a condition that induces triglyceride accumulation. The authors show that high fat diet feeding leads to increased CTPS mRNA levels, longer CTPS cytoophidia, and increased lipid droplet size in the fat body, a functional homolog of liver and adipose tissue. They go on to show that fat body-specific knockdown of CTPS or expression of mutant CTPS that disrupts cytoophidia formation blunts triglyceride storage in response to high fat and normal diets. Knockdown of CTPS reduces expression of lipogenic genes such as SREBP, ACC and FASN. Finally, the authors present evidence that insulin signaling is disrupted in fat body cells with impaired CTPS function. Altogether, the authors make a strong case for the requirement of CTPS for normal lipid storage. However, rescue experiments that could determine which cellular pathway(s) affected by CTPS knockdown are critical for lipid storage are lacking, raising the possibility that alternative processes affected by loss of CTPS, in particular cell growth, are in fact the primary defect.

1. Formation of CTPS cytoophidia promotes endoreplication in the *Drosophila* ovary and salivary gland (Wang et al., Genetics, 2015), and cell growth driven by myc requires CTPS (Aughey et al., PLOS Genetics, 2016). Given that larval fat body cells grow to large sizes via endoreplication, a process with high nucleotide demand, it is possible that fat body cell size may be reduced when CTPS is knocked down (as the images in Figures 5D and 5F suggest). Fat body cell and nuclear size should be quantified in cells expressing CTPS RNAi. This could be done using a clonal approach to eliminate potential effects of organ-wide CTPS dysfunction on the internal milieu. If cell size is indeed reduced by CTPS knockdown, this may reduce the capacity for triglyceride storage.

2. Reduction in insulin signaling in fat body cells with CTPS knockdown is suggested by reduced tGPH fluorescence intensity and decreased phospho-Akt signal in immunostaining experiments (Figures 5D-5F and 6E-6H). Does the reduced tGPH signal result from decreased protein expression or decreased localization to the plasma membrane where PIP3 is synthesized? Using a clonal approach to assess insulin signaling in individual fat body cells with CTPS knockdown would provide stronger evidence that this pathway is inhibited by loss of CTPS.

3. Adult flies with ubiquitous or fat body-specific knockdown of CTPS (presumably from the larval stage onward) exhibit increased sensitivity to starvation, but whether this is due to defective triglyceride storage (as suggested by data later in the paper) is untested. Do these flies have reduced triglyceride storage at the onset of starvation and/or changes in the rate of triglyceride breakdown during starvation?

4. The authors show that reduced triglyceride storage is a consequence of impaired CTPS function, and they find that expression of the master lipogenic regulator SREBP is reduced when CTPS is knocked down. Does expression of a wild type or N-terminal domain SREBP that mimics the cleaved, active form (SREBP 1-452) rescue triglyceride storage in larvae co-expressing CTPS in RNAi in fat body?

5. A strong possibility for decreased triglyceride storage in fat bodies lacking CTPS is that the phospholipids that require CTP for production are reduced, thereby limiting lipid droplet membrane expansion. Are phospholipid levels reduced in fat bodies lacking CTPS?

1. The authors should show the degree of knockdown with the various CTPS RNAi transgenes used. Are any of the genes encoding nucleotide diphosphate kinases, another cellular source of CTP, induced when CTPS is depleted?

2. In Figure 4A, cg-GAL4 is used to drive UAS-GFP with or without a CTPS RNAi transgene. Co-expression of CTPS RNAi strongly reduces GFP fluorescence. Why?

3. What is the viability of larvae raised on the 30% coconut oil diet? Do they survive to adulthood?

4. Has the anti-phospho Thr308 Akt antibody been validated in *Drosophila*? If not, its specificity should be tested in clones of fat body cells expressing a Pdk1 RNAi transgene. The authors should also determine total Akt levels in cells lacking CTPS.

5. The CTPS-mCherry allele has been generated for this manuscript. Additional information describing this allele – does it behave as wild type, where is the tag in the genome, etc – should be presented in a supplementary figure.

[Editors’ note: further revisions were suggested prior to acceptance, as described below.]

Thank you for resubmitting your work entitled "Fat body-specific reduction of CTPS alleviates HFD-induced obesity" for further consideration by *eLife*. Your revised article has been evaluated by David James (Senior Editor) and a Reviewing Editor.

The manuscript has been improved however, all reviewers agreed that some of the major issues previously raised were either not addressed, or not addressed satisfactorily. We want to emphasize that unless the authors are willing to experimentally address each and every one of these points, with all appropriate genetic controls and under strictly controlled developmental conditions, the manuscript will no longer be discussed or considered for publication at *eLife*.

The remaining issues that need to be addressed are outlined below:

1. Performing experiments under the proper conditions and controls to address concerns about developmental timing, growth, and their indirect/direct impact on the fat phenotypes

2. Perform the additional controls for the SREBP experiment.

3. Perform the requested clonal analysis to address autonomous effects on fat and/or growth.

4. Acknowledge and consider, in the discussion, that impaired nucleotide synthesis (a very likely outcome of loss of CTPS) could result in impaired cell growth. In fat body cells, this impairment could lead to restricted endoreplication and restricted cell growth, which would likely result in altered larval fat phenotypes.

*Reviewer #1 (Recommendations for the authors):*

In this reviewed version of Liu et al., the authors have addressed most concerns raised by the reviewers. However, some of the major concerns remain to be addressed. Namely, the relationship between differences in growth and /or development between controls and experiments, and how these affect lipid levels has yet not been fully addressed.

1. It was requested that the authors ensure experiments are done in larvae of the same developmental timing and food density (known variables known to directly change lipid levels). The authors attempted to address this issue by opting for approaches that are not the most standard in the field (number of eggs laid instead of the number of first instars hatched is an example). I think some of the differences in size/growth and fat could be influenced by not remaining differences in developmental timing and therefore make it still hard to evaluate the results. A good example is when we look at the differences in size (and fat, based on the transparency of the larvae) and of the distinct controls presented in figures 4A and 6A- these controls appear to be at distinct stages of development (anywhere between early L3 and late L3 stages), times when fat changes can be significant.

2-The clonal analysis, where the genetic autonomous effects on fat and/or growth could have been tested, in the presence of endogenous controls, was not performed.

3- The request for additional controls to be used for the normalization of the RT-qPCRs was not addressed. The authors circumvolved the issue and attempted to validate rp49 as a sole endogenous control in a different context, not directly on their samples. I think the rationale presented for the validation of rp49 is flawed: if we consider that growth is indeed changing, both actin and rp49 could change proportionally, normalizing rp49 to actin in the different conditions as shown would indeed result in no changes for rp49. The standard for normalization for qPCR analysis is using 3-4 housekeeping genes that are measured simultaneously with the desired targets.

*Reviewer #2 (Recommendations for the authors):*

The authors have been highly responsive to all reviewers' comments, and new data in the paper strengthen their overall model and provide a clearer picture of the fat body phenotype when CTPS is knocked down. In particular, the quantification of cell and nuclear size as well as insulin signaling help to define potential driver phenotypes for reduced triglyceride storage in cells with CTPS depletion.

1. How might decreased growth in cells with low CTPS levels impact triglyceride storage? Normal cell growth may be permissive for triglyceride storage. Indeed, endoreplication in the larval fat body is thought to drive the accumulation of biomass in this organ over the course of the third larval instar, and high-fat diet feeding in this study leads to significant increases in cell and nuclear size. The possibility that fat storage phenotypes may derive from impaired cell growth should be considered in the Discussion section.

2. In Figure 5, the authors test whether a truncated SREBP-Cdel transgene that encodes the transcription factor domain can rescue phenotypes in the CTPS-depleted larval fat body. Data in this figure is difficult to interpret because the SREBP-Cdel transgene was not tested on its own. For example, if cell, nuclear, or lipid droplet size (panel G) or triglyceride storage (panel H) was increased by overexpression of SREBP-Cdel alone, then the data from animals co-expressing CTPS-RNAi and SREBP-Cdel would not suggest a rescue.

3. The authors provide a rationale for the use of CG-GAL4-driven expression of GFP as an indicator of fat mass as well as an explanation for reduced GFP expression in fat bodies co-expressing CTPS.RNAi. The most robust test of the use of cg>GFP as a fat mass indicator would be to measure GFP protein or transcript levels in whole larvae and in the fat body in controls and animals expressing CTPS.RNAi. One would expect the fat body GFP expression levels to be the same between genotypes and that the whole body GFP expression would decrease in animals with CTPS knockdown in the fat body. A variety of other transgenes that drive increased or decreased fat body mass could be tested as proof of principle.

4. Throughout the Figure Legends, sample sizes are included inconsistently. For example, in Figure 1, sample sizes are included for panels A-C and G-I but not D-F. All legends should include sample sizes for each panel.

*Reviewer #3 (Recommendations for the authors):*

The authors have significantly improved the experiments and interpretations in the revised text. They sufficiently addressed the reviewer's comments on their initial submission. I do not have any additional experimental suggestions that would enhance their findings and conclusions.

However, I do think it is important the authors address differences in growth and development between controls and experimental conditions. Additionally, the text should be further edited for readability and the telling of the story. At times the results read like a list of experiments and there is shifting between different tenses (e.g., use of past tense throughout most of the manuscript, but then future-tense in line 186 "…we would like to investigate…").

[Editors’ note: further revisions were suggested prior to acceptance, as described below.]

Thank you for resubmitting your work entitled "Fat body-specific reduction of CTPS alleviates HFD-induced obesity" for further consideration by *eLife*. Your revised article has been evaluated by David James (Senior Editor) and a Reviewing Editor.

The manuscript has been improved but there are some remaining issues that need to be addressed, as outlined below:

*Reviewer #2 (Recommendations for the authors):*

This revised manuscript addresses my concerns. However, the authors need to make two changes related to the SREBP.Cdel transgene.

First, on lines 346-348, the authors write, "overexpressing the truncated form of SREBP, which is constitutively activated due to the lack of the C-terminal transcriptional activation domain". This is wrong. In fact, the C-terminal deletion of SREBP in use here retains the activation domain but lacks the transmembrane domain that leads to retention in the secretory pathway unless cleaved by SCAP. This must be corrected.

Second, on lines 350-352, the authors write, "it is worth noting that overexpression of SREBP alone did not significantly enhance adipocyte size and nuclear size, despite significantly increasing TAG accumulation". The data in Figure 5G clearly show that overexpression of SREBPCdel actually *reduces* cell and nuclear size. At a minimum, the text should be corrected to describe the data accurately. Interpretation of the findings would be appreciated.

On a related note, it would be helpful to the reader to know why the authors used ppl-GAL4 rather than cg-GAL4 in the SREBPCdel experiments.

*Reviewer #3 (Recommendations for the authors):*

In this revised version of Liu et al., the authors have addressed most of the concerns raised by the reviewers. However, there are still a few issues of concern that should be addressed as described below before publication consideration:

1. The clonal analysis provided in Figure 4, supplement 2 (requested from reviewer 1, point 2 of previous critique) is not convincing and the statement in lines 262-263 that the clones are considerably smaller is not appropriate without quantification. The image provided looks like a large mass of one adipocyte cell instead of multiple smaller cells. If these were individual cells, it is unclear why the phalloidin stain is not outlining individual cells. In the image provided, it looks like the clone (or clones) are dying cells. In addition, there should be quantification of the clone size relative to the wild-type neighbors (even for supplemental data).

2. The authors should comment as to why overexpression of SREBP.Cdel alone results in decreased adipocyte and nuclear size, while increasing triacylglycerol levels. The authors claim that "CTPS is involved in the cell-autonomous regulation of adipocyte growth by maintaining the activation of the PI3K-AKT-SREBP signaling pathway." However, since overexpression of activated SREBP does not rescue all of the defects observed, there must be alternative explanations. Furthermore, in Figure 5F-H, the panel should like SREBP as SREBP.Cdel to be more consistent with the actual truncated transgenic construct that was used.

3. The image shown in Figure 5C for tGPH is not convincing for the plasma membrane to cytosol intensity. In addition, it is unclear how the authors determined the plasma membrane for the cells in the middle of the image for the CTPS RNAi condition without an additional membrane marker. Furthermore, the methods provided for the parameters used for masking are lacking (lines 628-630).

---

## [Author Response]

[Editors’ note: the authors resubmitted a revised version of the paper for consideration. What follows is the authors’ response to the first round of review.]

Reviewer #1 (Recommendations for the authors):In this manuscript Liu and colleagues describe a role for the CTPS enzyme and cytoophidia with potentially functions in the fly fat body. The authors propose that CTPS and cytoophidia have a detrimental role in animals fed a high fat diet. The authors go on to propose that depletion of CTPS in HFD conditions reduces the number of cytoophidia formed and this results in a protective role against HFD-induced obesity. I find the work highly interesting and exciting, but somewhat preliminary. The target audiences would be those in the areas of metabolism and general cellular biology.At this stage, several issues require additional consideration, including: how changes in growth might contribute to the observed phenotypes; the specific effects of high-fat diet versus other diets; changes in lipids and phospholipids versus the direct effects of CTPS and cytoophidia. There are instances where controls for experiments are missing, compromising the ability to interpret the data as fully supporting the conclusions. Additionally, alternative approaches and/or conclusions have not been considered. Similarly, potential implications for such phenomena are not presented or discussed in the context of what is already known.

We would like to thank the reviewer for the insightful suggestions on our manuscript, which we have attempted to address as thoroughly as possible in our revision. We have specifically made the following changes:

The effects of CTPS deficiency on adipocyte growth have been investigated.We employed the RNAi control line (V60101 from the Vienna *Drosophila* Stock Center) or an overexpression control line (UAS GFP) as extra controls for body weight measurement, starvation assay, density assay, and TAG measurement. All data have been presented in the revised manuscript.We profiled the levels of phospholipids in the fat body expressing *CTPS*RNAi.To ensure that we appropriately discuss and clarify these issues, we extended the Discussion section to include the potential implications of such phenomena.

All data needs to be revised with more and better controls. Genetics and growth conditions play a strong influence in metabolic outcomes. Therefore, for assurance that some of the results are not just due to these but indeed to the experimental manipulations, the following major points need to be addressed:1.To rule out genetic contributions from insertions, each and all experiments need the genetic background control for each line, not just the drivers but also the RNA-i lines and UAS-overexpression lines. Each experiment also needs an overexpression control (GFP or similar) or a RNAi control (RFP-RNAi or similar). This is specifically requested for the all the life-span experiments, density assays and TAG measurements

Before genetic manipulation, the transgenic lines produced in the study had been backcrossed into the *w1118* background for more than five generations. This was noted in the revised methods (page 20, line 469). We performed the body weight measurement, starvation assay, density assay, and TAG measurement again, in which the RNAi control line (V60101 from the Vienna *Drosophila* Stock Center) or an overexpression control (UAS GFP) were introduced as extra control lines. We now present these new data in Figure 1A–C, 1G-I, 2A–B, 4A–B, 4E, 6A–C, and 6H of the revised manuscript.

2. Methods state that flies are allowed to lay for 4 hours, but don't state if density for each genotype in each vial was controlled for. Overcrowding or underpopulation of animals per vial/bottle can influence their ultimate size, food intake and metabolism. Were number of animals per vial controlled for? If not, please address.

To make sure the larvae used in this study were at the desired developmental stage, we restricted embryo collections by allowing females to lay eggs for less than 4 hours. Then around 80 embryos were placed into each vial of regular or high-fat food for the following experiments.

To perform body weight measurement and starvation assays, newly eclosed flies were collected and kept in standard fly food bottles (~200 flies per bottle) for 3–5 days. During this time, mating was unrestricted. For the starvation assay, male and female flies were then sorted into vials with 3 ml 1% agar, separately: 30 flies per vial, 5-6 groups per genotype. Flies were transferred every two days to avoid bacterial contamination. Every 12 hours, the number of dead flies was recorded as the time until death. We now note these in the revised manuscript’s methods (page 21, lines 493-499).

3. What are the levels of knockdown for each one of the RNAi lines? What are the levels of expression from each one of CTPS wt and mutant lines? Levels of knockdown and overexpression per driver could account for the differences in phenotype observed. Could you please show this is not the case?

Prompted by the reviewer’s suggestion, two *CTPS* RNAi lines that were employed in the study had their knockdown efficiencies assessed (Supplementary Figure 4A of the revised manuscript). The expression levels of *CTPS* in the *CgG4* > CTPS^WT^-OE and *CgG4* > CTPS^MU^-OE lines were also evaluated (Supplementary Figure 8A of the revised manuscript).

We agree fully with you that the driver determines the pattern and degree of expression of the manipulative gene. CTPS knockdown in diverse tissues resulted in varying degrees of body weight loss, which might be due to that GAL4 driver’s different pattern and strength. The knockdown efficiency of *CTPS* was then evaluated in various tissues using quantitative RT-PCR (Figure 1 DF of the revised manuscript). Lower knockdown efficiency of *CTPS* in the global (44%) than in the fat body (64%) (Figure 1D, F of the revised manuscript) may explain its weaker body weight loss in *TubG4^ts^* > *CTPS*-RNAi flies. A significantly higher knockdown level of *CTPS*-RNAi in pan neuron (82%) than in the fat body (Figure 1E, F of the revised manuscript), however, did not result in a reduction in body weight, demonstrating that CTPS in the fat body appears to be necessary for body weight increase. This has been included in the results on page 6, lines 121-130.

4. For evaluation of AKT activity via phosphorylation, an independent and quantifiable western blot for the fat bodies shown in Figure 5F,G and Figure 6G,H is requested.

In response to the reviewer's suggestion, we performed a western blot and assessed the independent Akt phosphorylation level and total Akt level in fat bodies using FIJI-ImageJ (Figure 5E and Figure 6H of the revised manuscript).

5. tGPH was developed to be used to measure insulin signaling by taking into account the amount of PIP3 in the nucleus versus the membrane. Total fluorescence is not commonly used as a reporter for INR signaling activity. Please either use the reporter as it was developed to be used, or demonstrate that total fluorescence can be used as an accurate reporter.

We apologize for the confusion. We employed tGPH as a reporter of PI3K activity in this investigation. For the purpose of comparing tGPH signals, we quantified the tGPH intensity ratio of the cell membrane to the cytosol in individual cells (Figure 5C, D, Figure 6F, and G of the revised manuscript). After manual Costes thresholding, fluorescence intensity was calculated using Imaris' intensity/voxel function. This was noted in the updated manuscript's results page 13, lines 298-299 and methods page 25, lines 576-580.

6. for figures 3A, 4A,C 5D,F, 6 D,E and G, some of these images seem to represent a more apical view of fat bodies than others. In order to compare levels of lipid droplets, tGPH, etc, images should be of a comparable central focal plane/z section, otherwise, lipid droplets can seem bigger or smaller, and nuclei fainter or brighter. Could you provide similar sections for each image?

Prompted by the reviewer’s suggestion, we obtained these images again. For comparison, the new image is that of the biggest nucleus in the central focal/z sector. This was mentioned in the methods on page 24, lines 565-566. The updated manuscript's Figure 3B-B’’, C-C’’, Figure 4C, F, Figure 5C, F, Figure 6D, and F display new images.

7. In figure 4, it seems that larval size changes with the CTPS-RNAi conditions. This would lead to different outcomes in terms of lipid levels and fat body sizes. Could you please measure larval size so that size as a variable is taken into account?

In response to the reviewer’s suggestion, we compared the larval size of the *CgG4* > *CTPS*-RNAi line to that of the control larvae. Although the *CgG4* > *CTPS*-RNAi larva was somewhat shorter, there was no statistically significant difference in size from the control larvae (Supplementary Figure 3A and B of the revised manuscript). This has been included in the results on page 8, lines 174-175.

8. Please use additional, and more than one, endogenous controls for the qPCR, as there seems to be a size difference in the RNAi larvae (above). RP49 is a ribosomal protein that correlates with cell size, and thus is not appropriate as the qPCR endogenous control.

Thank you for this suggestion. Using *actin* as the endogenous control, we carried out qRT-PCR to determine the expression levels of *rp49* and *GAPDH* (Author response image 1). While the *GAPDH* expression level significantly decreased in comparison to the control line, there was no discernible difference in the expression level of *rp49* between *CgGAL4* > *CTPS*-RNAi and the control line. *Rp49* is therefore a suitable choice for the endogenous control in this study.

**Author response image 1. sa2fig1:** Quantitative RT-PCR analysis (A) Quantitative RT-PCR analysis of *rp49* and *GAPDH* mRNA abundance in the fat body lysates of the third instar larvae from the indicated genotypes (30 larvae/genotype; 3 groups/genotype, 3 biological replicates). All data are shown as mean ± S.E.M. ns, no significance, ** P < 0.01, by two-way ANOVA with a Tukey *post hoc* test.

9. In figure 4, why are levels of GFP lower in the RNAi condition? This is not expected, as the GFP should still be labelling the fat cells that are present.

We employed *CgG4* in combination with UAS-eGFP as an indication of fat mass (Figure 4A, A’, and A’’ of the revised manuscript). Following HFD feeding, wild-type larval eGFP fluorescence intensity was increased, demonstrating an expansion in adipose tissue mass. Importantly, as compared to larvae under RD conditions, the fluorescence intensity of HFD-fed *CgG4*, eGFP > *CTPS*RNAi larvae was not substantially increased (Figure 4A’’ of the revised manuscript). When the fat body was dissected, we noticed that the total amount of fat body from HFD-fed *CgG4*, eGFP > *CTPS*-RNAi larva was significantly less than that of *CgG4*, eGFP > + larva (Figure 4A’ of the revised manuscript), which provides us an explanation for the noticeably decreased eGFP intensity in the *CgG4*, eGFP > *CTPS*-RNAi line (Figure 4A, A’’ of the revised manuscript). This has been included in the results on pages 9-10, lines 205-215.

10. In the Discussion, please discuss what is known about CTPS and cytoophidia in the context of lipid metabolism and growth regulation. What role do the authors imagine in the fly fat body and in response to diet? Is this role expected to be high-fat-diet-specific? What would be the role of the cytoophidia? And how would this role lead to regulation of the expression of lipogenic genes?

To ensure that we appropriately discuss and clarify these issues, we extended the Discussion section to include these contents. Please refer to the discussion on pages 16-19, lines 376-448.

11. CTPS regulation of lipid metabolism and phospholipids: do the observed effects result from accumulation of CTPS products, or due to the formation of cytoophidia per se?

Despite the fact that species differ in their ability to form filaments, the structurebased studies show that CTPS filaments dynamically switch between active and inactive states in response to variations in substrate and product levels (Lynch EM *et al.*, *Nat Struct Mol Biol.* 2020; Zhou X *et al.*, *JGG.* 2019). This filament-based mechanism of improved cooperativity shows how CTPS polymerization may be adapted to produce different regulatory consequences. Our study has shown that filament formation increases protein stability and enzymatic activity (Sun Z and Liu JL, *Cell Discov.* 2019). Therefore, we could not exclude the possibility that product depletion, such as that of CTP, a product conversing from UTP enzymatically catalyzed by CTPS, contributed to reduced lipogenesis when CTPS was knocked down in fat bodies. The reduction in HFDinduced adipocyte expansion and TAG level in the absence of cytoophidia, whether resulting to reduced CTPS expression or the expression of a mutant CTPS protein that acts as a "dominant-negative" protein to prevent the polymerization, strongly suggests that cytoophidia play an important role in lipid metabolism. This has been included in the discussion on pages 16-17, lines 376-397 and page 18, lines 420-426.

Reviewer #2 (Recommendations for the authors):In this manuscript Liu et al. determine that CTP synthase (CTPS) is required in the fat body to regulate TAG levels and body weight. The authors also show that CTPS transcript levels are more abundant on a high fat diet and propose that manipulation of CTPS specifically in the fat body rescues defects observed. The authors perform RNA sequencing of fat bodies from control and CTPS RNAi larvae and determine that genes involved in lipogenesis are significantly decreased. The authors also provide fluorescence analysis that CTPS may interact with the PI3K pathway to regulate lipid levels. The authors suggest that CTPS cytoophidia regulate induction of high fat diet related obesity.The initial observation that CTPS regulates lipid levels in the fat body through regulation of PI3K signaling. However, it is unclear how the authors came to some of their conclusions and how their results significantly enhance findings from prior studies characterizing this protein and the cytoophidia that form.1. It is unclear whether transgene induction occurred during development for their analysis of adults (Figure 1). Also, did the authors maintain the flies for RNAi at 25 degrees or shift them to 29 degrees?

In this study, we employed the *tublin-GAL4, tublin-GAL80^ts^* line as a ubiquitous driver, the *Elav-GAL4* line as a pan-neuron driver, and the *Cg-GAL4* and *ppl-GAL4* lines as fat body drivers. Prompted by the reviewer, the knockdown efficiency of *CTPS* in different tissues was evaluated, and the expression level of *CTPS* was reduced by 44% in the global, 82% in the pan neuron, and 64% in the fat body, respectively (Figure 1 D-F of the revised manuscript). In all experiments, flies were kept at 25 °C. This is now noted in the revised methods on page 21, lines 499-500.

2. Could the authors provide reasoning as to why global knockdown of CTPS using tubulin-Gal4 gives a relatively weak phenotype in body weight compared to fat body-specific RNAi (Figure 1A and C)?

When compared to fat body-specific knockdown, *tubulin-GAL4* exhibits a comparatively mild phenotype in terms of body weight (Figure 1A and C of the original version of the study). The *tubulin GAL4* flies we initially employed were lost during the pandemic. Another driver, *tublin-GAL4, tublin-GAL80^ts^* line utilized in the modified version also showed a relatively weak phenotype compared to *Cg-GAL4* (Figure 1A, C of the revised manuscript). CTPS knockdown in diverse tissues resulted in varying degrees of body weight loss, which might be due to that GAL4 driver’s different pattern and strength. The knockdown efficiency of *CTPS* in various tissues was then assessed using qRTPCR (Figure 1 D-F of the revised manuscript). The weaker body weight loss in *TubG4^ts^*>*CTPS*-RNAi flies might be explained by a lower knockdown efficiency in the global (44%) than in the fat body (64%) (Figure 1D, F of the revised manuscript). This has been included in the results on page 6, lines121-131.

3. Why do the authors propose that there is a decrease in starvation resistance when CTPS is knocked down in the central nervous system using Elav-Gal4, but not a change in body weight (Figure 1B and E)?

Prompted by the reviewer, in the revised manuscript, we employed *ElavG4* > Control-RNAi and *CTPS*-RNAi/+ as additional controls. In comparison to the *ElavG4* > + and *ElavG4* > Con-RNAi flies, the *ElavG4* > *CTPS*-RNAi fly still showed dramatic defects in survival during food deprivation, with reductions in their median survival rates of 20% and 11% for females and 26.7% and 26.7% for males, respectively. Whereas, the *ElavG4* > *CTPS*-RNAi flies didn’t show a significant reduction in body weight or deficiency in starvation resistance as compared to CTPS-RNAi/+ flies (Figure 1B, H of the revised manuscript). The female *ElavG4* > *CTPS*-RNAi flies even have a longer starved duration than female *CTPS*-RNAi/+ flies (Figure 1H of the revised manuscript).

4. The authors should address the fact that the Cg-Gal4 driver used in their study is shut off by knockdown of CTPS in both a regular and high fat diet. For example, using Cg-Gal4 in combination with UAS-GFP is an indication of driver activity. The authors should provide reassurance to the readers that they are getting sufficient knockdown of CTPS by qRT-PCR and provide some explanation as to how they can make comparisons/conclusions between regular and high fat diets. This manipulation can alter the conclusions made in their studies going further using this set up. It may be beneficial to do a time course to determine how long after transgene induction that this switch occurs. Alternatively, the authors could determine if PPL-Gal4 would be a more suitable driver for these experiments. Furthermore, the authors should ensure that the driver remains active in their overexpression analysis with wild-type and the H355A mutant.

Prompted by the reviewer, two *CTPS* RNAi lines that were employed in the study had their knockdown effectiveness assessed by qRT-PCR (Supplementary Figure 4A of the revised manuscript), indicating that the *Cg-GAL4* driver (*CgG4*) was not shut off by knockdown of *CTPS*. When the fat body was dissected, we noticed that the total amount of fat body from HFD-fed *CgG4*, eGFP > *CTPS*-RNAi larva was significantly lower than that of HFD-fed *CgG4*, eGFP > + larva (Figure 4A’ of the revised manuscript), explaining why the eGFP intensity was noticeably lower in the *CgG4*, eGFP > *CTPS*-RNAi line (Figure 4A, A’’ of the revised manuscript). This has been noted in the results on pages 9-10, lines 205-215.

We also evaluated the expression levels of *CTPS* in the *CgG4* > CTPS^WT^ and *CgG4* > CTPS^H355A^ lines by qRT-PCR (Supplementary Figure 8 of the revised manuscript).

5. The authors propose that the GFP level fluorescence increase using the Cg-Gal4 driver on a high fat diet suggests increased lipogenesis (line 171). However, this observation is not quantified and there is not any evidence that the Cg-Gal4 driver can be used as a readout of lipid levels. This driver just represents a regulatory region between the Collagen IV genes (vkg and Cg25C – see Asha et al. 2003).

We agree that the *CgG4* driver can’t be employed as a lipid level readout. We employed *CgG4* in combination with UAS-eGFP as an indication of fat mass. We are very grateful for this reviewer identifying the inappropriate statement and bringing it to our attention. The sentence “an increase in fat storage suggesting that HFD-induced elevate lipogenesis.” was removed from the revised manuscript. Additionally, the eGFP fluorescent intensity was quantified (Figure 4 A’’ of the revised manuscript).

6. How is body weight gain determined (line 179-180)?

We added the formula for body weight gain to the revised manuscript’s methods on page 22, lines 509-510. The body weight gain was calculated using the following formula:Δ BW: Body weight gainΔ BW=BWHFD−BWRDReduction of body weight =(Δ BWCon−Δ BWCTPS−Ri)Δ BWConx 100%

7. Some of the images provided in the manuscript do not look like the fluorescence intensity has been scaled between the control and experimental images equivalently. For example, in Figure 4C the regular diet control and CTPS RNAi lipid droplet intensity does not look the same. Even if there are fewer and smaller, the BODIPY staining intensity in general should be uniform.

Thank you for your suggestions. We obtained the images again. For comparison, the new image is that of the biggest nucleus in the central focal/z sector. This was mentioned in the methods section (page 24, lines 563-566). The updated manuscript's Figure 3B, C, Figure 4C, F, Figure 5C, F, Figure 6D, and F display new images.

8. The authors propose that CTPS regulates PI3K signaling and show activity using reporters and antibody. Given the number of tools in the *Drosophila* community, this study would be greatly enhanced if genetic interaction experiments were performed.

Thank you for this great suggestion. We carried out the genetic rescue experiment. We found that defective adipocyte expansion, smaller lipid size, and reduced lipid accumulation caused by *CTPS* knockdown could be at least partially rescued by overexpressing the active form of SREBP (Figure 5F, G, and H of the revised manuscript).

9. The authors should change loss of CTPS inhibits SREBP transcriptional activity (lines 221-222) to inhibits SREBP transcript levels.

Corrected (page 12, line 282).

10. It is unclear why there is an increase in body weight with overexpression of wild-type CTPS, but not an increase in TAG levels (Figure 6B). Could the authors please provide an explanation?

In the original version, the TAG level was normalized to the body weight resulting in a lower relative TAG level due to the higher body weight gain of CTPS^WT^-OE. In the revised manuscript, we employed *CgG4* > UAS-GFP and CTPS-RNAi/+ as additional control lines in response to the reviewer’s advice. We repeated the body weight and TAG tests. To get more pronounced lipogenesis effects, the experiment's high-fat diet recipe was slightly modified in the current study. Additionally, the TAG level presented in the current study was normalized to the protein level rather than body weight. The new data on body weight and TAG levels in CTPS^WT^-OE and CTPS^MU^-OE are now presented in Figure 6B and C of the revised manuscript. The updated recipe for a high-fat diet was provided in the revised methods section (page 21, lines 490-492). Furthermore, overexpression of CTPS^MU^ still led to reduced body weight and TAG level in the current study, which was consistent with the original results (Figure 6B and C of the revised manuscript).

11. The authors claim that lipid droplet accumulation and size are changed with overexpression of WT and the H355A CTPS mutant, but do not provide any quantification (Figure 6D).

The lipid droplet size in fat bodies from overexpression of CTPS^WT^ and CTPS^H355A^ larvae were quantified. The data are now provided in Figure 6E of the revised manuscript.

12. The authors should provide the full recipe for their regular food diet as there is not a "standard yeast-cornmeal-agar food" (line 380).

We included the full recipe for the regular diet in the revised manuscript’s methods section (page 21, lines 488-490).

13. What is PBSTG (line 427)? Please provide the recipe.

The PBSTG recipe is included in the updated methods section (page 23 line 552-553).

14. The manuscript should be further edited for clarity. For example, there are quite a few run-on sentences that are difficult to follow. In addition, summary statements/conclusions of the data in the Results section (as done for lines 189-190) would be helpful to the reader.

The manuscript has been edited. We modified the Results section of the revised text to include the data summary statements and conclusions.

Reviewer #3 (Recommendations for the authors):This work investigates the role of CTP synthase (CTPS) in the control of triglyceride storage in response to normal and high fat diet feeding in *Drosophila melanogaster*. CTPS generates the nucleotide CTP, a molecule required for phospholipid, RNA and DNA synthesis. CTPS is regulated, in part, by its ability to form filaments called cytoophidia that promote enzyme activity. Using loss-of-function approaches in Drosophila, the authors find that CTPS is required for growth and lipid accumulation in flies in normal conditions and in response to high fat diet feeding, a condition that induces triglyceride accumulation. The authors show that high fat diet feeding leads to increased CTPS mRNA levels, longer CTPS cytoophidia, and increased lipid droplet size in the fat body, a functional homolog of liver and adipose tissue. They go on to show that fat body-specific knockdown of CTPS or expression of mutant CTPS that disrupts cytoophidia formation blunts triglyceride storage in response to high fat and normal diets. Knockdown of CTPS reduces expression of lipogenic genes such as SREBP, ACC and FASN. Finally, the authors present evidence that insulin signaling is disrupted in fat body cells with impaired CTPS function. Altogether, the authors make a strong case for the requirement of CTPS for normal lipid storage. However, rescue experiments that could determine which cellular pathway(s) affected by CTPS knockdown are critical for lipid storage are lacking, raising the possibility that alternative processes affected by loss of CTPS, in particular cell growth, are in fact the primary defect.

We sincerely appreciate the reviewer’s informative comments on our manuscript, and we have endeavored to address them as fully as possible in our revision. Specifically, we have made the following revisions:

1. We conducted rescue experiments that demonstrated CTPS knockdown suppresses the PI3K-AKT-SREBP1 pathway, which is required for lipid accumulation and adipocyte expansion.

2. We examined the effects of CTPS deficiency on adipocyte growth in the revised manuscript.

1. Formation of CTPS cytoophidia promotes endoreplication in the *Drosophila* ovary and salivary gland (Wang et al., Genetics, 2015), and cell growth driven by myc requires CTPS (Aughey et al., PLOS Genetics, 2016). Given that larval fat body cells grow to large sizes via endoreplication, a process with high nucleotide demand, it is possible that fat body cell size may be reduced when CTPS is knocked down (as the images in Figures 5D and 5F suggest). Fat body cell and nuclear size should be quantified in cells expressing CTPS RNAi. This could be done using a clonal approach to eliminate potential effects of organ-wide CTPS dysfunction on the internal milieu. If cell size is indeed reduced by CTPS knockdown, this may reduce the capacity for triglyceride storage.

Prompted by the reviewer’s suggestion, we obtained the adipocyte images again. For comparison, the new image is that of the biggest nucleus in the central focal/z sector. This was mentioned in the methods section (page 24, lines 563-566). The nuclear and adipocyte sizes of the larval fat body were quantified (Figure 4F, G, and H of the revised manuscript). The results showed that *CTPS* knockdown decreased adipocyte size under RD and HFD conditions. Unfortunately, we were unable to perform clonal experiments due to lack of reagents.

2. Reduction in insulin signaling in fat body cells with CTPS knockdown is suggested by reduced tGPH fluorescence intensity and decreased phospho-Akt signal in immunostaining experiments (Figures 5D-5F and 6E-6H). Does the reduced tGPH signal result from decreased protein expression or decreased localization to the plasma membrane where PIP3 is synthesized? Using a clonal approach to assess insulin signaling in individual fat body cells with CTPS knockdown would provide stronger evidence that this pathway is inhibited by loss of CTPS.

We determined the expression level of four subunits of *Pi3K* in the fat body using qRT-PCR in order to ascertain if lower protein expression of *Pi3K* is the cause of the diminished tGPH signal. *Pi3K* expression levels did not significantly decrease (Supplementary Figure 7A of the revised manuscript) when CTPS was knocked down, suggesting that the decreased tGPH signal may be caused by lower PI3K activity that decreases tGPH localization to the cell membrane. The phosphorylation of Akt was reduced (Figure 5E of the revised manuscript). Furthermore, we performed the rescue experiment and found that overexpression of the active form of SREBP could at least partially repair the lower lipid accumulation, smaller lipid size, and defective adipocyte expansion induced by *CTPS* knockdown (Figure 5F, G, and H of the revised manuscript). Together, the results show that CTPS is required for PI3K-AktSREBP signaling to remain activated.

3. Adult flies with ubiquitous or fat body-specific knockdown of CTPS (presumably from the larval stage onward) exhibit increased sensitivity to starvation, but whether this is due to defective triglyceride storage (as suggested by data later in the paper) is untested. Do these flies have reduced triglyceride storage at the onset of starvation and/or changes in the rate of triglyceride breakdown during starvation?

Prompted by the reviewer’s suggestion, we investigated TAG levels in male adults to determine whether the starvation sensitivity of *CgG4* > *CTPS*-RNAi flies is due to a shortage of lipid storage. TAG content in *CgG4* > *CTPS*-RNAi flies was reduced by 74.1%, 83.5%, and 62.5% under fed conditions when compared to *CgG4* > +, *CTPS*-RNAi/+, and *CgG4* > Con-RNAi flies, respectively (Supplementary Figure 2A of the revised manuscript). TAG levels in flies steadily decreased when they were starved. The TAG in *CgG4* > *CTPS*-RNAi flies was almost completely broken down over a 24-hour period of food restriction, explaining their lower median survival rate in comparison to the control lines (Figure 1I of the revised manuscript). Therefore, we proposed that part of the sensitivity in starvation resistance that we observed may be explained by insufficient TAG storage when CTPS was deficient. This has been included in the Results section (pages 7-8, lines 157-167).

4. The authors show that reduced triglyceride storage is a consequence of impaired CTPS function, and they find that expression of the master lipogenic regulator SREBP is reduced when CTPS is knocked down. Does expression of a wild type or N-terminal domain SREBP that mimics the cleaved, active form (SREBP 1-452) rescue triglyceride storage in larvae co-expressing CTPS in RNAi in fat body?

We are very grateful for this reviewer’s great advice. We carried out the rescue experiments. We found that overexpression of the active form of SREBP at least partially rescued the reduced adipocyte size and smaller lipid droplet size caused by *CTPS* deficiency (Figure 5F and G of the revised manuscript). Additionally, in SREBP-overexpressing larvae, the TAG content was also partially restored (Figure 5H of the revised manuscript). This has been included in the Results section (pages 13-14, lines 313-319).

5. A strong possibility for decreased triglyceride storage in fat bodies lacking CTPS is that the phospholipids that require CTP for production are reduced, thereby limiting lipid droplet membrane expansion. Are phospholipid levels reduced in fat bodies lacking CTPS?

Prompted by the reviewer’s suggestion, we profiled the levels of phospholipids in the fat body expressing *CTPS-RNAi*. The level of the main phospholipids, including phosphatidylethanolamine (PE), phosphatidylcholine (PC), lysophosphatidylethanolamine (LPE), and lysophosphatidylcholine (LPC), when normalized to protein concentration, slightly decreased although not significantly (Supplementary Figure 5A of the revised manuscript). CTPS deficiency led to modestly reduced phospholipid biosynthesis, which could result in smaller lipid droplets in adipocytes. This has been included in the Results section (pages 10-11, lines 238-244).

6. The authors should show the degree of knockdown with the various CTPS RNAi transgenes used. Are any of the genes encoding nucleotide diphosphate kinases, another cellular source of CTP, induced when CTPS is depleted?

Prompted by the reviewer, two *CTPS* RNAi lines that were employed in the study had their knockdown efficiencies assessed by qRT-PCR. The knockdown efficiencies of *CTPS* from the two RNAi lines were comparable (Supplementary Figure 4A of the revised manuscript). Additionally, we also assessed the expression levels of genes such as *nmydn-D6*, *nmydn-D7*, and *CG15547*, which encode the nucleotide diphosphate kinases that catalyze the conversion of CDP to CTP. We found that the expression of *nmydn-D7* was augmented in fat bodies expressing *CTPS*-RNAi (Supplementary Figure 5B of the revised manuscript), indicating that the fat body may enhance the expression level of the nucleotide diphosphate kinase to compensate for CTP production when de novo CTP synthesis is inhibited. This has been included in the Results section (page 11, lines 246-253).

7. In Figure 4A, cg-GAL4 is used to drive UAS-GFP with or without a CTPS RNAi transgene. Co-expression of CTPS RNAi strongly reduces GFP fluorescence. Why?

We used *CgG4* in combination with UAS-eGFP as an indication of fat mass (Figure 4A, A’, and A’’ of the revised manuscript). With HFD feeding, wildtype larval eGFP fluorescence intensity increased, demonstrating an expansion in adipose tissue mass. Importantly, the fluorescence intensity of HFD-fed *CgG4*, eGFP > *CTPS*-RNAi larvae was not significantly increased when compared to HFD-fed *CgG4*, eGFP >+ larvae (Figure 4A’’ of the revised manuscript). When the fat body was dissected, we noticed that the total amount of fat body from HFD-fed *CgG4*, eGFP > *CTPS*-RNAi larva was significantly less than that of HFD-fed *CgG4*, eGFP > + larva (Figure 4A’ of the revised manuscript), which provides us an explanation for the significantly decreased eGFP intensity in the *CgG4*, eGFP > *CTPS*-RNAi line that we observed (Figure 4A, A’’ of the revised manuscript). This has been included in the Results section (pages 9-10, lines 205-215).

8. What is the viability of larvae raised on the 30% coconut oil diet? Do they survive to adulthood?

Larvae reared on a diet of 30% coconut oil had an ~ 80% viability rate. They can live to be adults. We found that newly enclosed flies can readily be trapped to death because HFD is sticky.

9. Has the anti-phospho Thr308 Akt antibody been validated in *Drosophila*? If not, its specificity should be tested in clones of fat body cells expressing a Pdk1 RNAi transgene. The authors should also determine total Akt levels in cells lacking CTPS.

Commercial Akt antibodies from Cell Signaling were utilized in the study and in other *Drosophila* experiments (Zhao P., *et al.*, *iScience*, 2021; Ding *et al.*, *Cell Reports*, 2021). In response to the reviewer’s suggestion, we performed an immunoblot examination of the levels of total and phosphorylated Akt in fat bodies expressing *CTPS*-RNAi, CTPS^WT^, and CTPS^MU^. The data are now presented in the updated manuscript's Figures 5E and 6H.

10. The CTPS-mCherry allele has been generated for this manuscript. Additional information describing this allele – does it behave as wild type, where is the tag in the genome, etc – should be presented in a supplementary figure.

We developed a "knock-in" fly line in which the coding sequences for the fluorescent protein mCherry and the V5 tag were inserted in-frame at the Cterminus of *CTPS* in order to observe the subcellular localization and dynamics of endogenous CTPS in vivo. We used immunofluorescence microscopy to directly identify CTPS protein in the fat body of the wild-type *w1118* fly line in order to ascertain whether the cytoophidium localization and shape seen in *CTPS*-mCh were an artifact induced by protein fusion between CTPS and mCherry or the V5 tag. The CTPS antibody revealed the presence of cytoophidia at the adipocyte cortex, supporting the findings from the CTPSmCh knock-in line (Supplemental Figure 1A in Liu J *et al.*, *Cellular and Molecular Life Sciences*, 2022). The details of *CTPS*-mCherry allele generation were disclosed in our recently published research (Liu J *et al.*, *Cellular and Molecular Life Sciences*, 2022). In the updated manuscript, we referenced it as Reference #25.

[Editors’ note: what follows is the authors’ response to the second round of review.]

The manuscript has been improved however, all reviewers agreed that some of the major issues previously raised were either not addressed, or not addressed satisfactorily. We want to emphasize that unless the authors are willing to experimentally address each and every one of these points, with all appropriate genetic controls and under strictly controlled developmental conditions, the manuscript will no longer be discussed or considered for publication at eLife.The remaining issues that need to be addressed are outlined below:1. Performing experiments under the proper conditions and controls to address concerns about developmental timing, growth, and their indirect/direct impact on the fat phenotypes2. Perform the additional controls for the SREBP experiment.3. Perform the requested clonal analysis to address autonomous effects on fat and/or growth.4. Acknowledge and consider, in the discussion, that impaired nucleotide synthesis (a very likely outcome of loss of CTPS) could result in impaired cell growth. In fat body cells, this impairment could lead to restricted endoreplication and restricted cell growth, which would likely result in altered larval fat phenotypes.

We deeply appreciate the valuable feedback provided by the editor and reviewers on our manuscript, and we have made every effort to address their concerns thoroughly in our revised version. Specifically, we have implemented the following revisions:

1. We ensured that our experiments were conducted with appropriate controls and conditions to elucidate the impact of CTPS on lipid metabolism and adipocyte growth.

2. We have performed additional controls for the rescue experiment that confirmed the role of CTPS in regulating the PI3K-AKT-SREBP pathway. All data have been presented in the revised manuscript.

3. We have conducted the clonal analysis in the revised manuscript to investigate the autonomous effects of CTPS on adipocyte growth.

4. We agree with the reviewer’s point that impaired nucleotide synthesis resulting from the loss of CTPS could lead to impaired cell growth, particularly in fat body cells. This could affect endoreplication and, ultimately, lead to altered larval fat phenotypes. We extended the Discussion section to include these contents in the revised manuscript.

We hope that these revisions have adequately addressed the reviewer's concerns and improved the quality of our manuscript.

Reviewer #1 (Recommendations for the authors):In this reviewed version of Liu et al., the authors have addressed most concerns raised by the reviewers. However, some of the major concerns remain to be addressed. Namely, the relationship between differences in growth and /or development between controls and experiments, and how these affect lipid levels has yet not been fully addressed.1. It was requested that the authors ensure experiments are done in larvae of the same developmental timing and food density (known variables known to directly change lipid levels). The authors attempted to address this issue by opting for approaches that are not the most standard in the field (number of eggs laid instead of the number of first instars hatched is an example). I think some of the differences in size/growth and fat could be influenced by not remaining differences in developmental timing and therefore make it still hard to evaluate the results. A good example is when we look at the differences in size (and fat, based on the transparency of the larvae) and of the distinct controls presented in figures 4A and 6A- these controls appear to be at distinct stages of development (anywhere between early L3 and late L3 stages), times when fat changes can be significant.

To minimize developmental timing variability among larvae in the study, we implemented three restrictions: limiting the time for egg collection to less than 4 hours, controlling larval density on the medium, and collecting larvae for the assay at a specific developmental stage (i.e 76 hours after egg laying, AEL). This ensured that the larvae used in the assay had a developmental timing variability within a 4-hour range. We have included these details in the Results section (Page 11, line 218-220), methods section and legends of the updated manuscript to provide more clarity. We compared differences in body weight, TAG assay, adipocyte size/growth, fat content, and biochemical assays (Western blot and qRT-PCR) between genotypes under RD or HFD conditions, using early third instar larvae (76~80-hour AEL larvae) in the study. This approach allowed us to evaluate and compare different phenotypes between the genotypes more accurately, under different conditions (as presented in Figure 4 and Figure 6 of the revised manuscript). To ensure precision and accuracy in our language, we have replaced the term "early third instar larvae" with the more specific "**-hour AEL larvae” (i.e, "76~80-hour AEL larvae") in the updated manuscript.

We acknowledge the reviewer's concern regarding the discrepancy in larval size between Figure 4A and 6A. We would like to clarify that the difference arose because live larvae were presented in the original Figure 6A while euthanized larvae, which were made rigid before imaging, were used in Figure 4A. As a result, the curvature of the live larvae during imaging could have affected their body size appearance, which may have caused the observed difference in morphology. In response to the reviewer's feedback, we have included a photograph of the euthanized larvae in the revised Figure 6A of the updated manuscript.

To ensure consistent larval densities between the control and experimental lines in our study, we used a method of transferring a precise number of eggs (i.e., 80 eggs) instead of first instar larvae into the medium. We chose this method because the experimental lines (*CgG4*/*CTPS*-Ri, *CgG4*/CTPS^WT^-OE, and *CgG4*/CTPS^MU^-OE) did not exhibit any defects in egg hatchability (as shown in Author response image 2), and therefore, the same number of eggs corresponded to similar larval density. In addition, transferring a precise number of eggs into a new food medium is a more efficient method and ensures that the larvae are undisturbed during feeding under either food condition, as compared to transferring first instar larvae. We appreciate the reviewer's valuable feedback, which has allowed us to improve the clarity of our manuscript.

**Author response image 2. sa2fig2:** Egg hatchability. (A-C) Egg hatching rate in the indicative lines under HFD condition (n=100 embryos/group, 3 groups/genotype, 2 biological replicates). All data are shown as mean ± S.E.M. ns, no significance by one-way ANOVA with a Tukey *post hoc* test.

2-The clonal analysis, where the genetic autonomous effects on fat and/or growth could have been tested, in the presence of endogenous controls, was not performed.

In response to the reviewer's suggestion, we performed clonal analysis in the fat body to provide evidence for the cell-autonomous regulation of CTPS. By crossing the *yw*, *hs-flp*; *act>CD2>G4,* UAS-GFP line with the *CTPS*-Ri line and inducing CTPS knockdown by heat shock, we generated CTPS-deficient fat body cell clones. As depicted in Figure 4—figure supplement 2A of the updated manuscript, the clones that carried CTPS RNAi were significantly smaller in size than their control counterparts. We have included these findings in the Results section on page 12, lines 257-263 of the updated manuscript.

3- The request for additional controls to be used for the normalization of the RT-qPCRs was not addressed. The authors circumvolved the issue and attempted to validate rp49 as a sole endogenous control in a different context, not directly on their samples. I think the rationale presented for the validation of rp49 is flawed: if we consider that growth is indeed changing, both actin and rp49 could change proportionally, normalizing rp49 to actin in the different conditions as shown would indeed result in no changes for rp49. The standard for normalization for qPCR analysis is using 3-4 housekeeping genes that are measured simultaneously with the desired targets.

Following the reviewer's suggestion, we utilized multiple endogenous control genes, including *actin*, *rp49*, and *rp32*, simultaneously with the target genes in the qRT-PCRs to enhance the reliability of our results. We have included the additional RT-PCR analyses using *actin* or *rp32* as internal controls in Figure1—figure supplement 1A, B, C, Figure5—figure supplement 2A, B and Figure6—figure supplement 2A of the revised manuscript.

Reviewer #2 (Recommendations for the authors):The authors have been highly responsive to all reviewers' comments, and new data in the paper strengthen their overall model and provide a clearer picture of the fat body phenotype when CTPS is knocked down. In particular, the quantification of cell and nuclear size as well as insulin signaling help to define potential driver phenotypes for reduced triglyceride storage in cells with CTPS depletion.1. How might decreased growth in cells with low CTPS levels impact triglyceride storage? Normal cell growth may be permissive for triglyceride storage. Indeed, endoreplication in the larval fat body is thought to drive the accumulation of biomass in this organ over the course of the third larval instar, and high-fat diet feeding in this study leads to significant increases in cell and nuclear size. The possibility that fat storage phenotypes may derive from impaired cell growth should be considered in the Discussion section.

We agree with the reviewer's insight that the absence of CTPS may cause impaired nucleotide synthesis, which could potentially affect cell growth, especially in fat body cells, ultimately leading to altered larval fat phenotypes. Thank you for highlighting this possibility, which may have contributed to the observed results, and for helping to improve the accuracy and thoroughness of our research.

We extended the Discussion section to include these contents on Page 21-22, line 476-483 in the revised manuscript.

2. In Figure 5, the authors test whether a truncated SREBP-Cdel transgene that encodes the transcription factor domain can rescue phenotypes in the CTPS-depleted larval fat body. Data in this figure is difficult to interpret because the SREBP-Cdel transgene was not tested on its own. For example, if cell, nuclear, or lipid droplet size (panel G) or triglyceride storage (panel H) was increased by overexpression of SREBP-Cdel alone, then the data from animals co-expressing CTPS-RNAi and SREBP-Cdel would not suggest a rescue.

In response to the reviewer's suggestion, we conducted additional controls for the rescue experiment. We found that overexpression of the active form of SREBP was able to partially restore the reduced lipid accumulation, smaller lipid size, and defective adipocyte expansion caused by *CTPS* knockdown.

(Figure 5F and G of the revised manuscript). However, it is worth noting that overexpression of SREBP alone did not significantly enhance adipocyte size and nuclear size (Figure 5F and G of the revised manuscript), despite significantly increasing TAG accumulation (Figure 5H of the revised manuscript). We have included all of these data in the updated Figure 5 F, G, and H in the revised manuscript.

3. The authors provide a rationale for the use of CG-GAL4-driven expression of GFP as an indicator of fat mass as well as an explanation for reduced GFP expression in fat bodies co-expressing CTPS.RNAi. The most robust test of the use of cg>GFP as a fat mass indicator would be to measure GFP protein or transcript levels in whole larvae and in the fat body in controls and animals expressing CTPS.RNAi. One would expect the fat body GFP expression levels to be the same between genotypes and that the whole body GFP expression would decrease in animals with CTPS knockdown in the fat body. A variety of other transgenes that drive increased or decreased fat body mass could be tested as proof of principle.

In this study, we used *CgG4* in combination with UAS-eGFP to indicate fat mass. In response to the reviewer's suggestion, we conducted tests on eGFP transcripts in the whole larvae and fat body in animals. The results demonstrated a significant reduction in eGFP transcripts of the whole body in the *CgG4*, eGFP > *CTPS*-Ri larvae, while no apparent change was observed in the fat body compared to the *CgG4*, eGFP > + larvae (Figure 4—figure supplement 1A of the revised manuscript). We have presented the data in Figure 4—figure supplement 1A of the revised manuscript. We also investigated other transgenes, such as Akt and Myc,that are well known for their positive effects on fat body mass.

We found that knockdown of Akt, or Mys in fat body led to significant reduction in eGFP intensity (Author response image 3).

**Author response image 3. sa2fig3:** *CgG4* in combination with UAS-eGFP indicates fat mass. In comparison to the wild-type control, the third instar wandering larvae that were alive and expressed eGFP (green) exhibited either Akt or Myc knockdown in the fat body with *CgG4* driving. The dashed lines represent the extent of the larval bodies. The quantification of eGFP intensity is presented on the right panel, and the value is normalized to the control line *CgG4*, eGFP>+ (5 images/genotype, 3 biological replicates). The data are presented as mean ± S.E.M. No significance was observed (ns), and the statistical analysis showed **** P < 0.0001 by using a two-way ANOVA with a Tukey post hoc test.

4. Throughout the Figure Legends, sample sizes are included inconsistently. For example, in Figure 1, sample sizes are included for panels A-C and G-I but not D-F. All legends should include sample sizes for each panel.

Corrected.

Reviewer #3 (Recommendations for the authors):The authors have significantly improved the experiments and interpretations in the revised text. They sufficiently addressed the reviewer's comments on their initial submission. I do not have any additional experimental suggestions that would enhance their findings and conclusions.However, I do think it is important the authors address differences in growth and development between controls and experimental conditions. Additionally, the text should be further edited for readability and the telling of the story. At times the results read like a list of experiments and there is shifting between different tenses (e.g., use of past tense throughout most of the manuscript, but then future-tense in line 186 "…we would like to investigate…").

We sincerely appreciate the reviewer’s valuable comments on our manuscript. In order to minimize developmental timing variability among larvae used in the study, we implemented three restrictions: limiting the time for egg collection to less than 4 hours, controlling larval density on the medium, and collecting larvae for the assay at a specific developmental stage (i.e 76 hours after egg laying, AEL). These ensured that the larvae used in the assay had a developmental timing variability within a 4-hour range. These approaches allowed us to evaluate and compare different phenotypes between the genotypes more accurately, under different conditions. We have included these details in the Results section (Page 11, line 218-220), methods section and legends of the updated manuscript to provide more clarity. To ensure precision and accuracy in our language, we have replaced the term "early third instar larvae" with the more specific "**-hour AEL larvae” (i.e, "76~80-hour AEL larvae") in the updated manuscript. Additionally, we have made further edition for the manuscript.

[Editors’ note: what follows is the authors’ response to the third round of review.]

The manuscript has been improved but there are some remaining issues that need to be addressed, as outlined below:Reviewer #2 (Recommendations for the authors):This revised manuscript addresses my concerns. However, the authors need to make two changes related to the SREBP.Cdel transgene.First, on lines 346-348, the authors write, "overexpressing the truncated form of SREBP, which is constitutively activated due to the lack of the C-terminal transcriptional activation domain". This is wrong. In fact, the C-terminal deletion of SREBP in use here retains the activation domain but lacks the transmembrane domain that leads to retention in the secretory pathway unless cleaved by SCAP. This must be corrected.

Corrected.

Second, on lines 350-352, the authors write, "it is worth noting that overexpression of SREBP alone did not significantly enhance adipocyte size and nuclear size, despite significantly increasing TAG accumulation". The data in Figure 5G clearly show that overexpression of SREBPCdel actually *reduces* cell and nuclear size. At a minimum, the text should be corrected to describe the data accurately. Interpretation of the findings would be appreciated.

Corrected.

We changed the “it is worth noting that overexpression of SREBP alone did not significantly enhance adipocyte size and nuclear size” to “it is worth noting that overexpression of SREBP.Cdel alone significantly reduced adipocyte size and nuclear size” on lines 356-358, p16 in the updated manuscript.

Interpreting phenotypes resulting from the constitutive overexpression of a key metabolic signaling player can be challenging. In the context of SREBP overexpression, it was observed that this transcription factor is capable of increasing lipogenesis without concomitantly increasing cell size. These findings suggest that the precise expression level of SREBP may play a crucial role in the regulation of cell size. In the revised manuscript, we have included this interpretation of the observations in the Result section (lines 358-364, p16).

On a related note, it would be helpful to the reader to know why the authors used ppl-GAL4 rather than cg-GAL4 in the SREBPCdel experiments.

In this study, we aimed to investigate the role of CTPS in the fat body by inducing CTPS-RNAi expression using two different GAL4 drivers, *Cg-Gal4* and *ppl-Gal4*. Both drivers led to a reduction in CTPS expression in the fat body, resulting in a similar metabolic phenotype. While both *Cg-Gal4* and *ppl-Gal4* can be used as drivers in the rescue experiment, we chose to utilize the pplGAL4 driver for experimental reasons.

Initially, we attempted to generate the *CgG4*; *CTPS*-Ri and the *pplG4*; *CTPS*Ri fly lines to carry out the rescue experiments. However, we encountered challenges with the *CgG4*; *CTPS*-Ri fly line as it exhibited significantly reduced vitality in adult flies, making it impractical to obtain healthy flies for further cross experiments.

Given the constraints with the *Cg-Gal4* driver line, we decided to proceed with the *ppl-Gal4* driver for the rescue experiments. This driver line displayed better viability and allowed us to conduct the necessary experiments effectively.

Reviewer #3 (Recommendations for the authors):In this revised version of Liu et al., the authors have addressed most of the concerns raised by the reviewers. However, there are still a few issues of concern that should be addressed as described below before publication consideration:1. The clonal analysis provided in Figure 4, supplement 2 (requested from reviewer 1, point 2 of previous critique) is not convincing and the statement in lines 262-263 that the clones are considerably smaller is not appropriate without quantification. The image provided looks like a large mass of one adipocyte cell instead of multiple smaller cells. If these were individual cells, it is unclear why the phalloidin stain is not outlining individual cells. In the image provided, it looks like the clone (or clones) are dying cells. In addition, there should be quantification of the clone size relative to the wild-type neighbors (even for supplemental data).

We thank the reviewer for the suggestion. We have performed the quantification of cell size and nuclear size in the clonal analysis. The results have been presented in the Figure 4—figure supplement 2 B and C of the revised manuscript. Additionally, we have included images with higher magnification (Figure 4—figure supplement 2A, i, and ii of the revised manuscript) to clearly visualize individual clone cells.

2. The authors should comment as to why overexpression of SREBP.Cdel alone results in decreased adipocyte and nuclear size, while increasing triacylglycerol levels. The authors claim that "CTPS is involved in the cell-autonomous regulation of adipocyte growth by maintaining the activation of the PI3K-AKT-SREBP signaling pathway." However, since overexpression of activated SREBP does not rescue all of the defects observed, there must be alternative explanations. Furthermore, in Figure 5F-H, the panel should like SREBP as SREBP.Cdel to be more consistent with the actual truncated transgenic construct that was used.

Interpreting phenotypes resulting from the constitutive overexpression of a key metabolic signaling player can be challenging. One possible explanation for the observed phenotypes is that cellular homeostasis is intricately balanced and controlled by multiple regulatory mechanisms. Overexpression of SREBP.Cdel may disrupt this delicate balance, leading to fitness costs that impact on adipocyte and nuclear size. Furthermore, adipocyte growth and lipid metabolism are governed by a complex network of interconnected pathways, where SREBP is just one component among many regulators involved in modulating these processes.

In our study, we found that overexpression of SREBP.Cdel partially rescued reduced cell and nuclear size and TAG level induced by *CTPS* deficiency. However, it is important to note that overexpression of SREBP.Cdel alone resulted in decreased adipocyte and nuclear size, despite increasing triacylglycerol levels. This observation offers a plausible explanation as to why overexpression of activated SREBP does not completely rescue all the defects caused by CTPS knockdown.

It suggests that although overexpression of SREBP promotes lipogenesis, it may not lead to a proportional increase in cell size. This emphasizes the critical role of maintaining precise levels of active SREBP for effective regulation of cell size. In the revised manuscript, we have included this interpretation of the observations in the Result section (lines 358-364, p16).

We have changed SREBP to SREBP.Cdel in Figure5F-H of the revised manuscript, as the reviewer’s suggestion.

3. The image shown in Figure 5C for tGPH is not convincing for the plasma membrane to cytosol intensity. In addition, it is unclear how the authors determined the plasma membrane for the cells in the middle of the image for the CTPS RNAi condition without an additional membrane marker. Furthermore, the methods provided for the parameters used for masking are lacking (lines 628-630).

We have addressed the reviewer's concerns in the revised manuscript. We have provided the higher magnification images in Figure 5C to provide clearer visualization of tGPH recruitment on the plasma membrane.

The updated Figure 5C now demonstrated that CTPS-Ri led to a noticeable decrease in tGPH recruitment on the membrane, and the cell outline in *CTPS*Ri was distinguishable. The methods for the parameters used for masking has provided in the Method section of the revised manuscript (lines635-644, p28).